# Land Use Change Scenario Building Combining Agricultural Development Policies, Landscape-Planning Approaches, and Ecosystem Service Assessment: A Case Study from the Campania Region (Italy)

Elena Cervelli [1,2,3], Pier Francesco Recchi [1], Ester Scotto di Perta [1] and Stefania Pindozzi [1,2,3,4,*]

1   Department of Agricultural Sciences, University of Naples Federico II, Via Università 100, 80055 Portici, Italy; elena.cervelli@unina.it (E.C.); pierfrancesco.recchi@unina.it (P.F.R.); ester.scottodiperta@unina.it (E.S.d.P.)
2   Interdepartmental Laboratory of Territorial Planning (LUPT), University of Naples Federico II, Via Toledo 402, 80134 Naples, Italy
3   Task Force on Smart and Sustainable Mobility SUM, University of Naples Federico II, 80134 Napoli, Italy
4   BAT Center—Interuniversity Center for Studies on Bioinspired Agro-Environmental Technology, University of Naples Federico II, 80055 Portici, Italy
*   Correspondence: stefania.pindozzi@unina.it; Tel.: +39-0812539128

**Abstract:** In the last two centuries, land-use change (LUC) has been the most important direct change driver for terrestrial ecosystems. In contrast with the consequent ecosystem degradation, forward-looking spatial policies and target landscape and land-use planning processes are needed from a sustainability perspective. The present paper proposes a framework of action, including different landscape-planning and ecological approaches: from spatial modelling to recognize LUC and build different scenarios, to ecosystem service (ES) assessment to evaluate possible environmental impacts. Three different scenarios were explored: Trend, No Tillage, and Energy crops. The sediment delivery ratio and carbon storage and sequestration ESs were assessed and compared for each scenario. The results show that regional development in line with past trends could lead to further land degradation (with ES value losses, in a decade, greater than 5%). Instead, the two scenarios proposed in compliance with EU policies could bring benefits, if only those related to moderate LUCs and respecting the naturally grass-vegetated land. The aim of the paper is to support decision makers and local communities in the landscape planning landscape planning process. From the local to global scale, guided and shared LUC management allows us to implement sustainable development, based not only on a deep knowledge of the physical environment but also of social and economic issues.

**Keywords:** land-use change; land-use planning; ecosystem services; erosion; climate change; agricultural policies; soil tillage

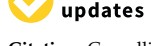



## 1. Introduction

Land-use change (LUC) is considered one of the most important drivers of environmental degradation in the last two centuries [1] and negative LUCs' effects on the environment are increasingly ascertainable: habitat endangerment, nutrient-cycle alteration, hydrogeological risk, etc. [2,3]. Guiding LUC, or at least mitigating its effects, is one of the global challenges that academia and international organizations, such as local communities, are facing. In the last decade, different landscape policies (at the international, community, and national level) have promoted sustainable land-use strategies relating to different interests: nature, agriculture, energy, infrastructure, housing, etc. At the European level, the Common Agricultural Policy (CAP) [4], the European Strategy for Biodiversity 2030 [5], the Green Deal [6], the Renewable Energy Directive I, II, III [7], etc., all recognize a fundamental role for landscape and its correct management in achieving sustainable development, although none of them indicate precisely how and where they might accomplish

their respective goals [8]. As a result, excessive soil consumption is still the main concern linked to the lack of coordination among policies [9]. Policy makers should be able to find a synthesis between human needs and the preservation of natural resources, considering local necessities and global issues [10].

A combination of different land-use and landscape-planning approaches can be useful to achieve sustainable development. The study of past LUCs and their implications is key to building future scenarios, which allow us to assess the potential impact of anthropic activities on the environment and human wellbeing. In this field, both land-use modelling [11,12] and the mapping of ecosystem services (ESs) [1,13,14] are valid tools supporting decisionmakers in their choices. Although the scientific community has recognized the need to integrate ES assessment into the landscape-planning process [15], the application of integrated evaluations is still not widespread; there is still a lack of knowledge that needs to be remedied, especially as regards the standardization of methods and facility of application in environmental impact assessments [16].

The aim of the present work is to propose a framework of actions which, integrating these different landscape/ecological planning approaches, supports decisionmakers and local communities to achieve sustainable development in the rural landscape.

The present paper, in compliance with the Drivers-Pressures-State-Impacts-Responses (DPSIR) model [15,17,18], builds different LUC scenarios according to alternative EU agricultural policies (CAP), in order to verify whether these policies are feasible at the local level and which implications they could have for the landscape. To this end, four different scenarios were compared in terms of expected ecosystem services. The study area is the Campania region (Southern Italy), which has witnessed significant LUCs in the last decades [19]. The present study focuses on rural areas, simulating two different kinds of LUCs associated with the new 2023–2027 Italian Common Agricultural Policies (CAPs) aimed at the mitigation of soil erosion, in comparison with the current trend of change and no changes option. Both scenarios are devoted to analyzing changes of arable land patterns, in one case as a consequence of a crop change to energy crops (to meet Green Deal requirements), in the other as a consequence of a cultivation-technique change from conventional to no-tillage.

Since LUC is not the only factor affecting ecosystem degradation, in the present study, the understanding of specific landscape-management implications was also deepened. Land management has a profound impact on the provision of ESs overall, since it can improve system efficiency and environmental quality [16,20,21]. Accordingly, different kinds of land management may determine different outputs of ESs, supported by new and more articulated assessment methods integrating expert opinions and spatial modelling tools [22–24]. Based on these premises, the proposed framework of action aims to deepen our understanding of whether, among agricultural uses, different management policies and strategies could directly influence not only human activities, but also the environmental impacts and ES supply, avoiding, or at least mitigating, the negative effects of LUCs.

As the study was conceived as cross-cutting research, with interactions and dialogues between intermediate and final results at every phase, this paper entails the following main steps:

- Scenario building: three different scenarios of land-use change in the Campania region (southern Italy) were built, in addition to the current state analysis.
- The selection of ESs, their quantification and the GIS-supported mapping.
- The comparison of scenarios in terms of possible LUC impacts.
- decision makers and communities' support, by means of specific comments on scenarios.

Specifically, the present work starts by building different scenarios, identifying the possible hot spots of LUC with the use of Dyna-CLUE modelling. Then, the work focuses on two specific ES assessments by means of InVEST software version 13.13.0, with specific attention to landscape use and management. The sediment-delivery ratio (SDR) and carbon storage and sequestration were the reference ecosystem services investigated in this study. This choice was driven by the desire to relate a typical regional-scale issue, such as soil erosion, to a global one, which is $CO_2$ emissions, as a key challenge of policy making.

This case study also provides a methodological reference for rational spatial planning, proposing a framework of action referring to integrated approaches to land use and landscape planning. Understanding the dynamics of landscape transformation from the local to global scale helps limit or at least mitigate its negative effects. Agricultural areas can play a fundamental role in land-use-change planning and landscape management, both in terms of provisioning and regulating functions. The international, EU, and national strategies which link LUC and environmental impacts are useful guides at the global level to pursue sustainable development, but have to be improved at local level to find concrete application solutions and to mitigate and integrate LUC impacts. In rural areas, degraded, marginal, or fringe areas can respond better than others to the challenges of resilient land development.

## 2. Materials and Methods

### 2.1. Study Area

Campania is a region of southern Italy (Figure 1), located on the Tyrrhenian shore of the peninsula. It covers about 13.600 sq km, with a population of almost 6 million people, more than half of which live within the metropolitan area of Naples. The morphology and landscapes of the region are considerably heterogeneous in its different parts, so much so that it can be divided into several sub-regions: broad plains, internal hilly areas and mountain ranges, sparse mountains, promontories, volcanoes, and three main islands. The population and most human activities are concentrated on the plains, especially the Volturno and Sele valleys, determining a vast, complex pattern of continuous and discontinuous urban fabric mixed with industrial areas and plots of intensive agricultural land [25]. The surrounding hilly areas are less populated and mainly shaped by extensive agriculture. Campania holds several important natural or semi-natural areas: broad-leaved forests, Mediterranean coniferous forests, grassland, maquis, wetlands, cliffs, and other bare rock formations. The importance of such natural sites is testified to by the institution of two national parks and several other legally protected areas: 108 Sites of Community Importance (SCIs) and 31 Special Protection Areas (SPAs) (https://www.mase.gov.it/pagina/schede-e-cartografie (accessed on 24 March 2023)).

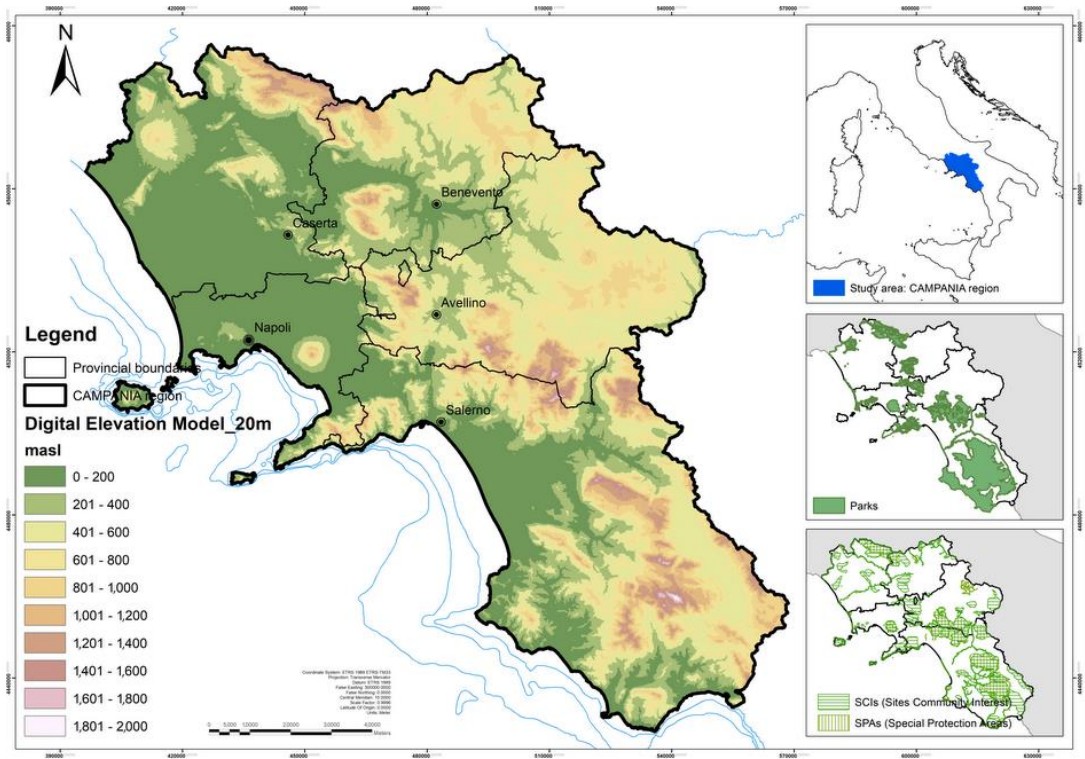

**Figure 1.** Study area. Campania region in southern Italy.

Although the primary sector accounts for a minor part of its GDP, Campania is one of the most important Italian regions in terms of agriculture, particularly regarding high quality agri-food products, and much of the territory has been deeply shaped by agriculture (https://land.copernicus.eu/pan-european/corine-land-cover (accessed on 24 March 2023)). The largest share of GDP is represented by services, among which tourism is the most important resource, thanks to a huge number of historical and archeological sites, such as Naples city center, Pompeii, seaside resorts, and world-famous beautiful landscapes, such as Sorrento, Capri, and the Amalfi Coast. In such a context, environmental recovery and landscape preservation are essential to regional development.

### 2.2. Steps in Method

Starting from the DPSIR framework (Figure 2), promoted by the Organization for Economic Cooperation and Development [26] to describe the interactions between a phenomenon and socioeconomic and environmental systems [27], the present work supports landscape planning and management in the Campania region, integrating different approaches focused on both land-use planning and landscape ecology.

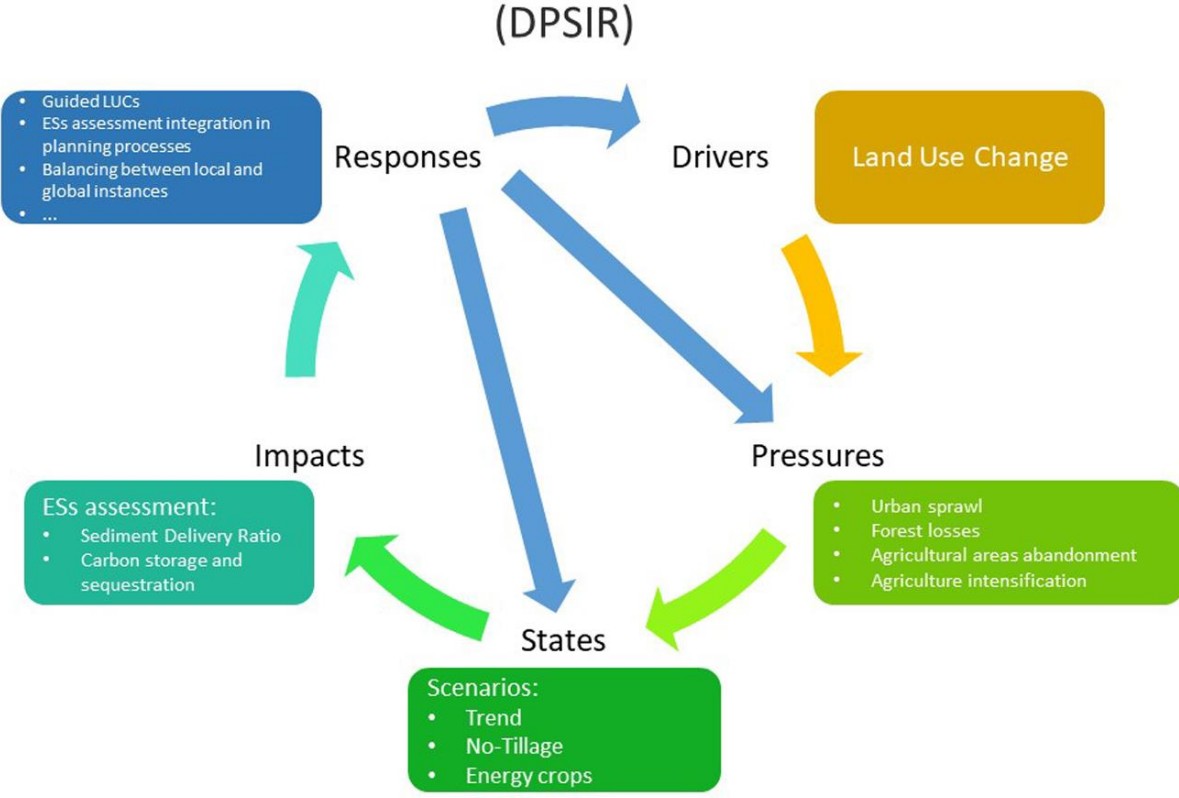

**Figure 2.** The DPSIR model and case-study implications.

The Dyna-CLUE model was used to recognize and identify past land-use changes and to build three possible LUC scenarios. InVEST software was used to assess and compare the possible impact of the hypothesized land-use changes on the environmental components. Considering the broad-scale level of analysis, Corine Land cover maps, referring to 2006, 2012, and 2018, were used. The spatial resolution of the study was fixed to the Corine Land Cover (1:100,000) in compliance with the correspondence with the ES classes found in the literature. Such spatial resolution does not allow for more detailed investigation [28], but it is useful as a preliminary assessment that can give interesting results consistent with



the scale usually addressed for strategic planning and, notably, for the decision-making support system. The methodology entailed three steps:

- Past LUC modelling analysis;
- Future LUC scenario building;
- ES assessment for each scenario.

*2.3. Land-Use-Change Scenario building*

The Campania region, since the sixties, has been subjected to substantial LUCs characterized by the complexity of local dynamics. In compliance with this past trend, it is possible to hypothesize that it is going to undergo an important LUC in the coming years [19,29], especially related to the depopulation of inland areas and the abandonment of agricultural uses inside these contexts, with consequent problems related not only to the loss of production and income, but also to landscape maintenance and care.

Future LUCs were simulated in three different scenarios developed under alternative strategies of land management.

- The "trend scenario" (TrS): starting from an awareness of past developments, a first scenario, resulting from the projection of the dynamics of the last 12 years, was developed;
- The "no-tillage scenario" (NTS): in compliance with CAP's aim, this scenario investigated the perspectives of cereal crops, as a consequence of European subsidies to farmers for no-tillage farming aimed at reducing soil erosion. Specifically, the scenario refers to the following measures: "ACA3—Reduced soil tillage techniques" Action 3.1—Adoption of no-till/"No tillage" (NT) seeding techniques"; and "Action 3.2—Adoption of minimal tillage/"Mini-mum tillage" (MT) and/or band working techniques/strip tillage";
- The "energy-crop scenario" (ECS): in compliance with RED I, II, and III policies, this scenario proposes the introduction of no-food energy crops as an alternative source of income for farmers and as a means of phytoremediation for polluted areas.

The overall duration of these simulations was 12 years, from 2018 to 2030, subdivided into yearly time steps.

A specific land-use map was developed and was used as the baseline for the comparison of scenarios, starting from the Corine Land Cover (CLC) maps referring to 2006, 2012, and 2018. The CLC maps, created within the European Program for Earth Observation, result from the processing of Landsat images with a nominal scale of 1:100.000, a minimum mapping unit of 25 ha, and a change-detection threshold of 5 ha. In these maps, different land-cover types are classified into 44 main classes on three levels of detail. In order to increase the thematic accuracy, which is currently about 85% [30], the CLC maps' precision was improved with information derived from LUCAS surveys [31] and aerial photographs. In the present paper, the different land-use types were merged into 8 thematic classes with a resolution of 100 m (Table S1).

The Dyna-CLUE (Dynamic Conversion of Land Use and its Effects) model was used for the simulation of the different scenarios [32]. This modelling system takes account of both intrinsic and extrinsic driving factors of LUC, so that different land-cover types are allocated at the grid-cell level in compliance with a weighted combination of location characteristics (geomorphology, climate, distance to main facilities and infrastructure, etc.), agents operating in the region, socioeconomic conditions, and spatial policies [32–35].

Agents' competitive strength plays an important role in the Dyna-CLUE modelling framework and it is weighted by setting a parameter known as 'conversion elasticity', which can assume values between 0 and 1 for the easy and difficult conversion, respectively, of a land-use type into different ones. The aptitude for conversion is related to the level of capital investment in each single class [36,37] so that the lowest value was given to non-irrigated arable land, while the highest value was given to urban fabric (Table S1).

In order to prevent urban areas from being converted into agricultural land, as well as all the other transitions that are actually quite unlikely to happen (Table S2), a conversion matrix representing the allowed changes between different land-cover types must be set [37].

The effect of location features on assigning each land-use type at grid-cell level is estimated by a regression analysis correlating every single land cover type, used as a dependent variable, with those factors deemed to be significant on its actual pattern, used as independent variables. Statistics are based on the logit model in compliance with the following formula (1):

$$\ln(Pi/1 - Pi) = \beta 0 + \beta 1 \times 1,i + \beta 2X2,i \ldots + \beta nXn,i \tag{1}$$

where Pi is the probability for the allocation of a specific land cover on the cell I; X1,2...n are the values assumed by the driving factors in the cell I; and β1,2...n coefficients are the weights of each factor in determining that land cover, calculated by running the regression, as well as the β0 constant value [37].

A stepwise procedure is used in the regression to exclude directly those variables not showing a significant influence on the actual land-use pattern [36].

Regression analysis was based on dependent variables derived from the 2012 land-cover map. The introduction of a new land-cover type such as energy crops, not yet existing in the current land-use classification, made it necessary to use a proxy variable to represent its hypothetical spatial distribution in compliance with a criterion of suitability [25,33]. Then, a suitability map based on Multi-Criteria Evaluation (MCE) was used to simulate a possible land-use pattern of energy crops in the related scenario. The energy-crop scenario map developed in Cervelli's [38] study of the Campania region was used. Geophysical and anthropic factors thought to be meaningful driving forces of LUC in the study area were used as independent variables of the regression analysis (Table 1).

**Table 1.** Geophysical and anthropic factors used as independent variables of the regression analysis.

| Sequence (As Used in Regression) | Factor (Driving Force) | Type of Variable |
|---|---|---|
| 1 | Elevation | Continuous |
| 2 | Slope | Continuous |
| 3 | Aspect | Continuous |
| 4 | Soil (Andosols) | Dummy |
| 5 | Soil (Cambisols) | Dummy |
| 6 | Soil (Luvisols) | Dummy |
| 7 | Soil (Calcisols) | Dummy |
| 8 | Soil (Vertisols) | Dummy |
| 9 | Distance to roads | Continuous |
| 10 | Distance to settlements | Continuous |
| 11 | Distance to streams | Continuous |
| 12 | Distance to coast | Continuous |
| 13 | Population density | Continuous |
| 14 | Temperatures | Continuous |
| 15 | Rainfall | Continuous |

In compliance with Pontius and Schneider [39], the validation of the regression analysis was performed by the relative operating characteristic (ROC) method, comparing the simulated spatial arrangement of every single land-use type with the actual 2018 one (or the suitability map as regards energy crops). ROC values higher than 0.5 suggest a correlation between predicted and observed land cover, the more significant the closer it is to 1.

The quantity of land undergoing conversion into different classes (demand) was calculated for every year of the simulation in compliance with the different purposes of each scenario [36,37].

Land requirements for TrS were computed in compliance with actual LUCs occurring during the period 2012–2018 in terms of absolute surfaces gained or lost by each land-cover class.

As regards NTS and ECS, demands are based on goals thought to be plausible, in compliance with the regional agricultural policies and in the wider context of current European agriculture conditions.

Specifically, regarding the NTS, it is related to the payments for environmental, climate, and other management commitments (ACA payments) that require very specific production and management behaviors of farmers. The ACA payments (which will operate very similarly to those of Measures 10 and 11 of the 2014–2020 programming), will have the objective of offsetting the higher costs and lost income associated with the voluntary adoption of the commitments for the climate and the environment. Among these ACAs, intervention "3" is intended for "ACA3—Reduced soil tillage techniques", as reported in the Italian National Strategic Plan, sent to the European Commission.

The ACA3 interventions, aimed at improving environmental performance, are divided into two actions (basic commitments):

- adoption of no-till/no-tillage (NT) seeding techniques;
- adoption of minimum-tillage techniques/minimum-tillage (MT) and/or band-tillage/ strip-tillage techniques.

Assuming that not all available soils will be maintained with no-tillage or minimum tillage techniques, an overall +2% was earmarked for non-irrigated arable land (in opposition to the actual negative trend) by the end of the simulation, mostly decreasing the actual positive trend of permanent crops. Regarding the other classes, they follow the trend, though at a lower rate than in the past.

Finally, regarding the ECS, in compliance with the same criteria, a new land-cover class of energy crops was introduced totaling about 40,000 ha, equal to 25% of the eligible areas pointed out by Cervelli [38], by the end of the simulation.

In Dyna-Clue model, the allocation of different land-use classes at grid cell level is performed by an iterative process, in compliance with location-based conditions, running until the total amount of LUCs meets the demand of each scenario, in a combination of a top-down and a bottom-up approach [32,40].

*2.4. The Ecosystem-Service Assessment*

The ES assessment within each scenario was developed using InVEST software (a system for the Integrated Valuation of Ecosystem Services and Tradeoffs), consisting in a set of models developed solely to quantify the provisioning of ESs as a result of land cover and land management [41]. The InVEST framework was developed by the Natural Capital Project, a partnership between Stanford University, The Nature Conservancy, and the World Wildlife Fund, with the specific aim of promoting ESs as guiding principles for decision making [42]. Unlike the common monetary methods, generally assigning flat values to each class of land cover in relation to a specific ES, InVEST models also take into account the effect of landscape configuration (e.g., edge effect, fragmentation, and distance to sources) on the provisioning of the ES [43]. As a result of modelling, ESs can be expressed both in biophysical and monetary terms, in accordance with the type of ES and the availability of data needed to run each model [41].

The ESs investigated in the present paper were the sediment-delivery ratio (SDR) and carbon storage and sequestration (CSS). The choice of these ESs arises from the need to verify the main impacts potentially linked to the hypothesized changes in land use. The three scenarios are closely related with the soil-erosion issue which, in many Mediterranean areas, represents the first step towards progressive desertification [44]. Soil erosion is mainly due to climate conditions, soil composition, and land management [45–50]. Since the first two factors are not directly manageable in compliance with human policies, land-management policies play a fundamental role in the mitigation of the phenomenon, encouraging or sustaining the incentivization of new crops in place of the current crops [51], as well as conservative agricultural practices like cover cropping and reduced tillage [21]. Particular attention was given also to soil carbon, considering the positive relation that it has with the erosion issue and in general with soil bio-physicochemical properties [52]. Moreover, the analysis of the landscape carbon stock was intended to relate a typically regional issue, erosion, with one of the greatest international issues, that is, global warming. In the present paper, the carbon storage and sequestration (CSS) service (InVEST software) was assessed

as a measure contributing to Earth's climate-change mitigation. Landscape management, as well as affecting an ecosystem's survival, plays an important role also in removing greenhouse gases (GHGs) such as $CO_2$ from the atmosphere; a landscape composed of forests, grasslands, peat swamps, and other terrestrial ecosystems collectively stores carbon [52,53], keeping $CO_2$ out of the atmosphere, where it would contribute to climate change.

2.4.1. The Erosion-Risk-Mitigation Ecosystem Service via the Sediment- delivery-ratio Model (InVEST Software)

The InVEST model dedicated to erosion is based on the universal soil-loss equation (USLE) [54], which is a widely acknowledged approach for estimating soil loss at watershed level. In compliance with this equation, the amount of soil loss (ton ha$^{-1}$ yr$^{-1}$) is given by the following formula (2):

$$USLE = R^* \times K \times LS \times C \times P \tag{2}$$

where R is the rainfall erosivity, K is soil erodibility, LS is the slope-length gradient factor, C is the cover-management factor and P is the support-practice factor. For a better understanding of these parameters, please refer to Wishmeier and Smith [54].

Not all the soil eroded generally reaches the catchment outlet, with the proportion that does depending on land morphology, which affects the shape and extent of the sediment stream. So, part of the sediment removed from an upslope location is held in a lower one, showing a certain retention effect. Hence, the InVEST model estimates the sediment export (ton pixel$^{-1}$ yr$^{-1}$) as follows (3):

$$E_i = USLE_i \times SDR_i \tag{3}$$

where $SDR_i$ is the sediment-delivery ratio, intended to be the proportion of soil loss in location i actually reaching the catchment outlet.

Since we worked at a resolution of 100 m, the final unit of measurement is ton ha$^{-1}$ yr$^{-1}$. The SDR calculation is based on the concept of a connectivity index illustrated by Borselli [55] and it relies on a high-resolution DEM as a function of the slopes.

Table 2 shows the data needed to run the model for every scenario. While R, K, and LS factors were considered to be constant over time, human-activity-dependent factors, such as land cover and management, were weighted in compliance with the different contexts investigated. As not enough data about anti-erosion support practices were available, the P-factor was assumed to be equal to 1.

**Table 2.** Sediment-delivery ratio—Input data and calibration parameters.

| Input Data | Type | Unit | Data Source |
|---|---|---|---|
| **DEM** | Raster map | m | DTM Digital Terrain Model 20 m (https://data.europa.eu/data/datasets/m_amte-299fn3 -eba41113-4141-4d46-9cdf-b0848deec44d?locale=it (accessed on 19 May 2022)) |
| **Rainfall erosivity index (R)** ® | Raster maps | MJ·mm·(ha·h·y)$^{-1}$ | The R factor was calculated, starting from 128 pluviometric stations in Campania region, in compliance with the Arnoldus equation and interpolated via the ArcGIS Kringing geostatistical analyst. |
| **Soil erodibility (K)** | Raster map | t·ha·h·(ha·MJ·mm)$^{-1}$ | The K factor was calculated in compliance with the Renard equation and interpolated via the ArcGIS Kringing geostatistical analyst tool. |
| **Land use/land cover (LULC)** | Raster map | Adimensional | The land-use maps were developed according the three scenarios via Dyna-CLUE model. |

**Table 2.** *Cont.*

| Input Data | Type | Unit | Data Source |
|---|---|---|---|
| **Watersheds** | Polygon shapefile | Adimensional | The watersheds were obtained starting from the DTM, via Hydrology tollset—Convert to shapefile. |
| **Biophysical table** | CSV table | Adimensional | The USLE "C" factor (coverage-management factor) was obtained starting from the LANDUM model (Section 2.4.2). The USLE "P" factor (support practice factor) was assumed to be equal to 1. |
| **Threshold flow accumulation (TFA)** | Number value | Adimensional | Derived from DTM via ArcGIS hydrology toolset. |
| **$k_b$ and $IC_0$** | Number value | Adimensional | The values used, derived from the literature, are $k_b = 2$ and $IC_0 = 0.5$ [56]. |
| **$SDR_{max}$** | Number value | Adimensional | The value used, derived from the literature, is 0.8 [57]. |

### 2.4.2. The C-Factor Deepening

In order to evaluate the impact on soil erosion of future LUCs driven by alternative strategies of land management, an estimation of the cover-management factor (C-factor) within the universal soil-loss equation (USLE) was carried out. The C-factor is a parameter ranging between 0 and 1 that takes into account the influence of land cover and management on soil loss, with bare soil having a C-factor value of 1 [54]. Despite the wide portion of the literature assigning uniform C-factor values in compliance with land cover [21,58,59], or remote sensing techniques [60–62], the present study attempted to consider the effect of different agricultural practices on arable lands, with special regard to conservative and no-tillage practices, as they are incentivized by CAP. The C-factor land-use and management model (LANDUM) [21] was therefore used to estimate a C-factor for each scenario. In compliance with LANDUM, the C-factor for arable lands is calculated as follows (4):

$$C_{arable} = C_{crop} \times C_{management} \tag{4}$$

where $C_{crop}$ is a constant depending on the crop type, while $C_{management}$ is a value depending on land-management practices.

The crop factor is calculated as a weighted average value based on the actual crop composition of arable lands in the region as follows (5):

$$Ccrop = \sum_{n=1}^{N} C_{cropn} \times S_{cropn} \tag{5}$$

where n is the crop type while S is its share at regional level.

Data about the crop composition of arable land in the Campania region are available on the Italian Institute of Statistics web portal [63] (ISTAT, 2016). Although crop rotation alters the pattern of arable lands quite quickly, crop composition can be considered stable in purely quantitative terms [21]; thus, it was approximated through an analysis over a period of six years, from 2006 up to 2011, years for which the most up-to-date information was available. In this way, it was possible to calculate the current $C_{crop}$, thought to stay the same up to 2030 in the TrS, while for the NTS and ECS, different coefficients were found in accordance with the supposed alterations in crop composition because of alternative policies.

The C-factor values taken as a reference in the present study were those suggested by Panagos [21].

The management factor is calculated as the result of different agricultural practices which affect soil loss as follows (6):

$$C_{management} = C_{tillage} \times C_{residues} \times C_{cover} \tag{6}$$

where $C_{tillage}$ is related to tillage practices, $C_{residues}$ to the maintenance of crop residues on the field, and $C_{cover}$ to eventual cover cropping.

Tillage practices, which represent one of the main issues of the present study, can be classified into three different types: conventional, conservative, and no-tillage (also known as zero tillage or sod seeding). Conventional tillage is meant as traditional preparation of soil for sowing by ploughing and harrowing; residues of previous crops are often harvested as byproducts. Conservative tillage is a type of land management which keeps a part of plant residues on the ground and, generally, adopts a lower level of soil processing, for instance not inverting the soil layers (vertical tillage), or at least choosing once-off ploughing. No-tillage farming involves sowing directly into the ground without any mechanical soil processing, in which case weed control is obtained through herbicides or mulching. As they can improve soil structure and stability, conservative and no-tillage practices represent an important strategy to reduce soil loss [16,64,65]. Among these different kinds of land management, untilled soils are those showing the lowest rate of erosion, ceteris paribus. In compliance with this assumption, the tillage factor is estimated by the following Equation (7):

$$C_{tillage} = 1 \times \%S_{conventional} + 0.35 \times \%S_{conservative} + 0.25 \times \%S_{notill} \tag{7}$$

where S is the surface treated, respectively, by conventional, conservative, and no-tillage practices.

Data about the share of tillage at the European, national, and regional level are available on the European Statistics Office web portal [66].

For TrS and ECS scenarios, the $C_{tillage}$ coefficients were considered to stay equal to the current value, since no reliable forecast can be made regarding the future share of arable land treated by different tillage practices.

For the NTS scenario, the achievement of 25% tilled surfaces has been hypothesized. Specifically, the conservative and the no-tillage scenarios were treated equally (12.5% + 12.5%).

Maintaining crop residues on fields and cover cropping, which consists in growing any kind of intermediate plantation with the specific aim of reducing soil loss during periods when land would be normally bare, can have a significant impact on soil protection [67,68]. As no data related to crop residues and cover crops are available at any scale, the $C_{residues}$ and $C_{cover}$ parameters were considered equal to 1.

The C-factor values for non-arable land were obtained from Diodato [58] and Fagnano [69] studies (Table 3).

**Table 3.** C-factor values for non-arable land.

| Land Cover | Further Detail | Literature C-Factor |
|---|---|---|
| Urban areas | - | 0.00 |
| Industrial areas and infrastructure | - | 0.00 |
| Arable land | Autumn-winter cereals | 0.20 |
| | Corn | 0.38 |
| | Legumes | 0.32 |
| | Potatoes | 0.34 |
| | Vegetables | 0.36 |
| | Oilseeds | 0.28 |
| | Fodder intercrops | 0.20 |
| | Meadow | 0.05 |
| | Fallow land | 0.50 |
| Permanent crops | - | 0.30 |
| Grassland and pastures | - | 0.05 |
| Forests | - | 0.03 |
| Energy crops * | - | 0.04 |
| Other land covers | - | 0.00 |

* Value was obtained from the "*A. donax*" cover class [69].

2.4.3. The Carbon-Sequestration Ecosystem Service via Carbon Storage and Sequestration Model (InVEST Software)

Global warming seems to be deeply connected with the increasing atmospheric concentration of greenhouse gases, among which carbon dioxide ($CO_2$) represents a serious concern, since a significant part of its emission is caused by human activities, in particular the combustion of fossil fuels and deforestation [70–72]. In the light of this, carbon sequestration is probably the most acknowledged ES [53]. The photosynthetic activity of organisms removes $CO_2$ from the atmosphere and stores it in biomass such as wood, which is one of the most important carbon stocks in terrestrial ecosystems. This is why LUCs, especially as regards the conversion of natural areas into agricultural or built-up land, play a key role in the carbon cycle [3]. Living beings are not the only carbon stock, but a considerable amount of organic matter is stored in the form of undecomposed dead material, humus, peat, and fossils, so much so that soil is actually the largest carbon stock of terrestrial ecosystems [73].

Land management, especially as regards crop rotation, fertilization, and tillage, has a deep impact on soil carbon stocks. Parenthetically, it should be noted that increasing soil organic content improves the stability of aggregates, helping mitigate erosion as well [16,20,52,74].

The InVEST model for mapping carbon sequestration calculates the total amount of elementary carbon stored in each grid cell, in terms of Mg ha$^{-1}$, as the sum of the four main carbon pools, that is, the aboveground biomass, belowground biomass, dead organic matter, and soil organic carbon. Thes carbon pools show different values, depending primarily on land cover but also on land management. In principle, forests show greater stocks in every pool. Grasslands also generally show high levels of carbon storage, while for agricultural land greater stocks of soil carbon can be found under a system of reduced tillage compared to conventional practices [74], all other things being equal. Each land-use class was therefore assigned a value for every single carbon pool estimated in accordance with the protocol proposed by the Intergovernmental Panel on Climate Change [75], which takes into consideration many different variables influencing carbon storage including the main species, latitude, land management and, as regards agricultural surfaces, even input level and tillage practices. Table 4 shows the carbon stocks of each land-cover type derived from IPCC reference values.

**Table 4.** Carbon stocks values for each land-cover type, derived from IPCC reference values.

| Land Cover | C Stock under 10% Reduced Tillage * [Mg/ha] | C Stock under 25% Reduced Tillage ** [Mg/ha] |
|---|---|---|
| Urban areas | 1.5 | 1.5 |
| Industrial areas and infrastructure | 8 | 8 |
| Non-irrigated arable land | 42 | 48 |
| Irrigated arable land | 42 | 42 |
| Permanent crops | 60 | 60 |
| Grassland and pastures | 86 | 86 |
| Forests | 243 | 243 |
| Energy crops | 62 | - |
| Other land covers | 0 | 0 |

* refers to the actual rate of reduced tillage; ** refers to a target rate driven by the incentivization of reduced tillage

As a result of modelling, a map of carbon storage was prepared out for each scenario. The amount of carbon sequestration can be calculated as the difference between the carbon storage at baseline and those estimated for future scenarios.

### 3. Results

*3.1. Regression Results and Model Validation*

Coefficients derived from logit model linking land-cover distribution to its driving factors in the Dyna-CLUE framework are shown in Table 5. The stepwise procedure used for the regression analysis excluded only one variable not showing a significant effect on landscape pattern: among the 15 variables considered, only the "distance to coast" variable seems not to have any influence on any land-cover type distribution, while all the others have shown significance at least to one class. The same table shows values of the ROC index for each one of the land-use classes. High ROC values were found. The higher value (ROC = 0.974) is related to urban areas. The lower value (ROC = 0.748) was found for industrial areas.

**Table 5.** Coefficients derived from logit model linking land-cover distribution to its driving factors in the Dyna-CLUE framework.

| Variable | Urban Areas | Industrial Areas and Infrastructure | Non-Irrigated Arable Land | Irrigated Arable Land | Permanent Crops | Grassland and Pastures | Forests | Energy Crops | Other Land Covers |
|---|---|---|---|---|---|---|---|---|---|
| Elevation | −0.001 | −0.004 | 0.001 | −0.006 | −0.001 | 0.002 | 0.001 | 0.004 | −0.002 |
| Slope | −0.031 | −0.096 | −0.175 | −1.360 | −0.011 | 0.039 | 0.079 | −0.056 | 0.022 |
| Aspect | n.s. | n.s. | n.s. | n.s. | n.s. | 0.001 | n.s. | n.s. | −0.003 |
| Soil (Andosols) | −0.322 | 0.932 | −0.581 | −0.941 | 1.569 | 0.479 | 0.223 | 1.762 | −2.308 |
| Soil (Cambisols) | n.s. | n.s. | −0.124 | 3.077 | 1.450 | 0.129 | −0.107 | 2.702 | −1.513 |
| Soil (Luvisols) | 0.115 | 0.578 | 0.160 | 0.576 | 1.469 | n.s. | −0.205 | 1.977 | −2.762 |
| Soil (Calcisols) | n.s. | n.s. | n.s. | n.s. | 1.841 | −0.457 | n.s. | 3.100 | −3.447 |
| Soil (Vertisols) | 0.222 | n.s. | 0.651 | n.s. | 0.959 | −0.662 | −0.471 | 3.546 | −3.633 |
| Distance to roads | −0.004 | n.s. | n.s. | n.s. | −0.001 | n.s. | n.s. | −0.001 | n.s. |
| Distance to settlements | −0.003 | 0.001 | n.s. | n.s. | n.s. | n.s. | n.s. | n.s. | n.s. |
| Distance to streams | n.s | n.s. | n.s. | n.s. | n.s. | n.s. | n.s. | n.s. | 0.001 |
| Distance to coast | n.s. | n.s. | n.s. | n.s. | n.s. | n.s. | n.s. | n.s. | n.s. |
| Population density | 722.533 | 96.455 | −1503.435 | −2214.155 | −997.370 | −1593.264 | −1612.075 | −954.241 | −760.251 |
| Temperatures | −0.258 | n.s. | 0.339 | n.s. | 0.209 | 0.279 | n.s. | 0.552 | n.s. |
| Rainfall | −0.001 | n.s. | −0.001 | −0.006 | n.s. | −0.001 | 0.001 | 0.001 | −0.001 |
| Constant ($\beta_0$) | 6.909 | −6.101 | −5.519 | 2.825 | −5.480 | −7.256 | −4.172 | −15.732 | −1.904 |
| **ROC** | **0.974** | **0.748** | **0.887** | **0.989** | **0.806** | **0.858** | **0.929** | **0.882** | **0.803** |

*3.2. The Land-Use-Change Scenario-building Results*

The overall amount of changes in each scenario (Figure 3) was estimated in compliance with its main driver, while their spatial configuration was simulated within the Dyna-CLUE framework. The results of modelling compared with the baseline of the current state of land use are shown in the next sections.

3.2.1. Past Changes and Trend Scenario (TrS)

The TrS scenario was developed in compliance with the LUCs that occurred in the reference period 2006–2018.

Concerning the past LUCs, the contingency matrix (Table 6) shows the net gain/loss of surfaces among the different LU/LC classes. Specifically, the table allows us to identify both the net gain/loss of surfaces, which are the most significant components of the overall change, while shifting surfaces had a lower impact on landscape arrangement, though they cannot be overlooked (for example, agricultural and natural areas).

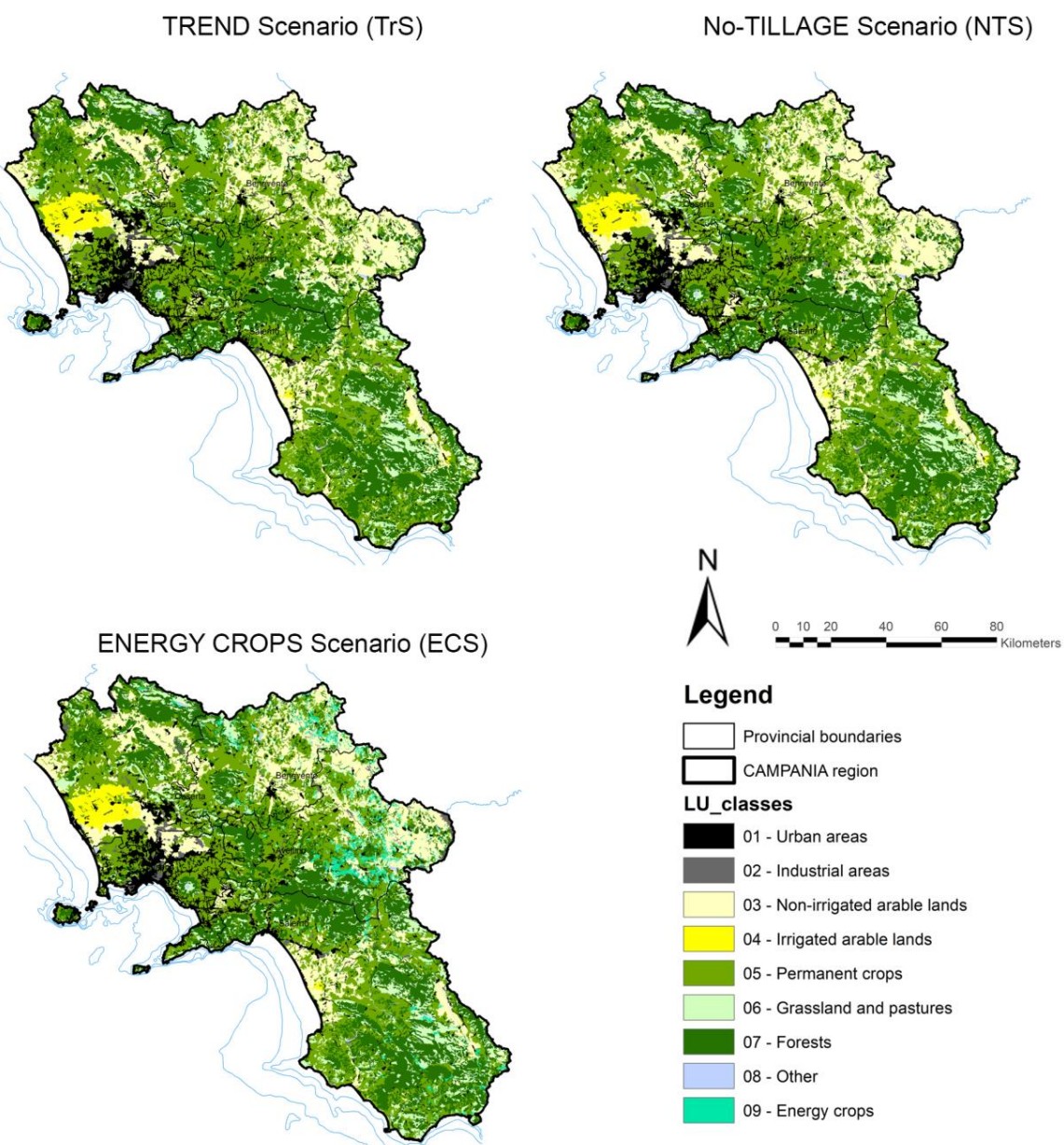

**Figure 3.** Land-use changes scenarios to 2030.

The same table shows the values of the ROC index for each one of the land-use classes. High ROC values were found. The higher value (ROC = 0.974) is related to urban areas. The lower value (ROC = 0.748) was found for industrial areas.

The urban sprawl can be considered the main driver of LUCs, especially regarding the increase in industrial areas (+58.3%), followed by that in urban ones (+13.2%), mostly at the expense of agricultural surfaces. Artificial surfaces tended to replace non-irrigated arable land and permanent crops, in most cases because of the expansion of existing settlements, which have a higher competitive strength for suburban locations.

Within the agricultural uses, non-irrigated arable lands are the only land-use type undergoing a significant net decrease (−4.7%), partly because of the aforementioned urban pressure, but mainly because they were converted into permanent crops, typically more profitable relative to their higher level of capital investment. As a matter of fact, 14,718 ha of non-irrigated arable land was replaced by permanent crops, which increased up to 3% in the reference period, while no significant changes affected irrigated arable land, which requires specific environmental characteristics (water availability, such as in riverside

flatlands). The ROC value associated with this land-use type is indeed the highest among the classes considered (0.989).

**Table 6.** Past LUCs (2006–2018)—Contingency matrix of land-use transitions. Rows are the land-cover-class composition in the year 2006; columns are the composition of the land-cover classes in the year 2018. The diagonals are the unchanged surfaces.

| Recognized LUC 2006/2018 | Urban Areas | Industrial Areas and Infrastructure | Non-Irrigated arable Land | Irrigated Arable land | Permanent Crops | Grassland and Pastures | Forests | Other Land Covers | TOTAL 2006 [ha] |
|---|---|---|---|---|---|---|---|---|---|
| Urban areas | **71,129** | 400 | 167 | 41 | 1027 | 11 | 42 | 0 | **72,817** |
| Industrial areas and infrastructure | 163 | **9221** | 62 | 0 | 161 | 58 | 4 | 283 | **9952** |
| Non-irrigated arable land | 1878 | 3319 | **281,059** | 592 | 14,718 | 718 | 257 | 172 | **302,713** |
| Irrigated arable land | 242 | 127 | 19 | **24,507** | 66 | 249 | 0 | 0 | **25,210** |
| Permanent crops | 8202 | 1578 | 3517 | 114 | **393,618** | 2163 | 2071 | 17 | **411,280** |
| Grassland and pastures | 399 | 976 | 3335 | 100 | 9640 | **131,381** | 4503 | 248 | **150,582** |
| Forests | 428 | 109 | 414 | 0 | 4260 | 3758 | **374,245** | 93 | **383,307** |
| Other land cover | 7 | 26 | 0 | 0 | 85 | 391 | 0 | **6082** | **6591** |
| TOTAL 2018 [ha] | **82,448** | **15,756** | **288,573** | **25,354** | **423,575** | **138,729** | **381,122** | **6895** | **1,362,452** |
| Net gain/loss [ha] | +9631 | +5804 | −14,140 | +144 | +12,295 | −11,853 | −2185 | +304 | 0 |
| Net gain/loss [%] | +13.2263 | +58.3199 | −4.6710 | +0.5712 | +2.9894 | −7.8714 | −0.5700 | +4.6123 | / |
| Simple annual rate [%] | +1.1021 | +4.8599 | −0.3892 | +0.0476 | +0.2491 | −0.6559 | −0.0475 | +0.3843 | / |

Similarly, the grasslands underwent a remarkable decline (−7.9%), mainly due to their replacement with non-irrigated arable land, permanent crops, and natural transition into forest.

No large changes affected forests, at least in terms of net gains or losses (−0.6%), although a small tradeoff between them and other land uses should be noted, especially permanent crops and grasslands.

This pattern endorsed the conversion-elasticity setting, which assigned lower values to those land-cover types showing a higher propensity to reversible changes [35–37].

Starting from the recognition of past LUCs, Table 7 shows a projection of land-use transitions up to 2030 resulting from the simulation. The LUC pattern is quite similar to the actual past one, with urban areas expanding at the expense of agricultural lands. These last in turn tend to shift to grass-vegetated unused land. Even within agricultural land, an analogous conversion of non-irrigated arable land into permanent crops can be seen.

There are only a few slight but evident differences between the observed and modeled changes.

The first is related to artificial surfaces, which were not allowed to be replaced by any other kind of land cover. In other words, urban expansion was set to be irreversible.

Another difference concerns the class of other land cover, which was set to stay the same for the duration of the simulation, as no reliable forecast can be made of fortuitous events or illegal actions, which can evidently be the only reasonable drivers of change within this category.

The last difference regards the missing tradeoff between forest and other non-artificial land, and even this is the result of a precise choice aimed at minimizing erroneous conversions. Since the transition of shrubs into forest is not location-based but rather age-based, no changes into forest were allowed because the period of simulation was considered too short to result in any appreciable natural reforestation.

**Table 7.** Trend scenario—TrS (2018–2030)—Contingency matrix of land-use transitions. Rows are the land-cover-class composition in the year 2018; columns are the composition of the land-cover classes in the year 2030. Diagonals are the unchanged surfaces.

| Simulated LUC 2018/2030 | Urban Areas | Industrial Areas and Infrastructure | Non-Irrigated Arable Land | Irrigated Arable Land | Permanent Crops | Grassland and Pastures | Forests | Other Land Covers | TOTAL 2018 [ha] |
|---|---|---|---|---|---|---|---|---|---|
| Urban areas | 82,448 | 0 | 0 | 0 | 0 | 0 | 0 | 0 | 82,448 |
| Industrial areas and infrastructure | 0 | 15,756 | 0 | 0 | 0 | 0 | 0 | 0 | 15,756 |
| Non-irrigated arable land | 9280 | 2877 | 262,897 | 3078 | 9235 | 1206 | 0 | 0 | 288,573 |
| Irrigated arable land | 382 | 190 | 1959 | 22,427 | 396 | 0 | 0 | 0 | 25,354 |
| Permanent crops | 960 | 63 | 3 | 0 | 422,546 | 3 | 0 | 0 | 423,575 |
| Grassland and pastures | 1841 | 4643 | 2021 | 499 | 10,176 | 119,549 | 0 | 0 | 138,729 |
| Forests | 1481 | 955 | 231 | 4 | 0 | 0 | 378,451 | 0 | 381,122 |
| Other land cover | 0 | 0 | 0 | 0 | 0 | 0 | 0 | 6895 | 6895 |
| **TOTAL 2030 [ha]** | **96,392** | **24,484** | **267,111** | **26,008** | **442,353** | **120,758** | **378,451** | **6895** | **1,362,452** |
| **Net gain/loss [ha]** | 13,944 | 8728 | −21,462 | 654 | 18,778 | −17,971 | −2671 | 0 | 0 |
| **Net gain/loss [%]** | +16.9124 | +55.3947 | −7.4372 | +2.5794 | +4.4332 | −12.9540 | −0.7008 | 0 | / |
| **Simple annual rate [%]** | +0.9395 | +3.0774 | −0.4131 | +0.1433 | +0.2462 | −0.7196 | −0.0389 | 0 | / |

### 3.2.2. No-Tillage Scenario (NTS)

Table 8 shows the results of modelling the NTS. This scenario is characterized by an expansion of agricultural land because of effective policies aimed at supporting no-tillage or minimum tillage practices. Unlike the TrS, non-irrigated arable land would undergo a significant increase (+2.1% by the end of the simulation), partly limiting the positive past trend of permanent crops. These last, in turn, will still increase, though at a lower rate than in the recent past (+1.3% by the end of the simulation).

**Table 8.** No-tillage scenario—NTS (2018–2030)—Contingency matrix of land-use transitions. Rows are the land-cover-class composition in the year 2018; columns are the composition of the land-cover classes in the year 2030. Diagonals are the unchanged surfaces.

| Simulated LUC 2018/2030 | Urban Areas | Industrial Areas and Infrastructure | Non-Irrigated Arable Land | Irrigated Arable Land | Permanent Crops | Grassland and Pastures | Forests | Other Land Covers | TOTAL 2018 [ha] |
|---|---|---|---|---|---|---|---|---|---|
| Urban areas | 82,448 | 0 | 0 | 0 | 0 | 0 | 0 | 0 | 82,448 |
| Industrial areas and infrastructure | 0 | 15,756 | 0 | 0 | 0 | 0 | 0 | 0 | 15,756 |
| Non-irrigated arable land | 4507 | 979 | 278,256 | 2342 | 1281 | 1208 | 0 | 0 | 288,573 |
| Irrigated arable land | 220 | 129 | 2113 | 22,503 | 389 | 0 | 0 | 0 | 25,354 |
| Permanent crops | 2453 | 74 | 13 | 90 | 420,942 | 3 | 0 | 0 | 423,575 |
| Grassland and pastures | 1651 | 6546 | 13,877 | 509 | 6568 | 109,578 | 0 | 0 | 138,729 |
| Forests | 1487 | 942 | 295 | 57 | 0 | 0 | 378,341 | 0 | 381,122 |
| Other land cover | 0 | 0 | 0 | 0 | 0 | 0 | 0 | 6895 | 6895 |
| **TOTAL 2030 [ha]** | **92,766** | **24,426** | **294,554** | **25,501** | **429,180** | **110,789** | **378,341** | **6895** | **1,362,452** |
| Net gain/loss [ha] | +10,318 | +8670 | +5981 | +147 | +5605 | −27,940 | −2781 | 0 | **0** |
| Net gain/loss [%] | +12.5145 | +55.0266 | +2.0726 | +0.5797 | +1.3232 | −20.1400 | −0.7296 | 0 | / |
| Simple annual rate [%] | +0.6952 | +3.0570 | +0.1151 | +0.0322 | +0.0735 | −1.1188 | −0.0405 | 0 | / |

Since this scenario is developed within a trend-based context, the effect of urban sprawl is still evident, with an overall increase of 12.5% and 55%, respectively, for urban and industrial areas, mainly at the expense of agricultural land.

Also, forests keep the same past trend with a little decline ($-0.7\%$ by the end of the simulation). Therefore, all the pressure of human activities would fall on grasslands, which would significantly decrease ($-20.1\%$ by the end of the simulation) mostly due to the expansion of non-irrigated arable land and permanent crops. In other words, even in this scenario, less competitive agriculture, in suburban locations, tends to shift onto non-used land.

In line with this trend, no important changes would affect irrigated arable land, whose highly specific sensibility to location characteristics makes it much more stable than other land-cover classes. Even in this case, the setting for allowed transition was kept as for the TrS (Table 7), considering the principles mentioned in Section 3.2.1.

### 3.2.3. Energy-crop scenario (ECS)

Table 9 shows the results in compliance with the energy-crop scenario. An overall expansion of agricultural land is determined by the introduction of energy crops for the bioremediation of polluted areas as proposed by Cervelli's [38] study. The ECS hypnotizes increasing the land used for energy crops up to about 40,000 ha, mostly competing with the non-irrigated arable lands and the grasslands, which are clearly the most lacking classes in terms of capital investment.

**Table 9.** Energy-crop scenario—ECS (2018–2030)—Contingency matrix of land-use transitions. Rows are the land-cover-class composition in the year 2018; columns are the composition of the land-cover classes in the year 2030. Diagonals are the unchanged surfaces.

| Simulated LUC 2018/2030 | Urban Areas | Industrial Areas and Infrastructure | Non-Irrigated Arable Land | Irrigated Arable Land | Permanent Crops | Grassland and Pastures | Forests | Other Land Covers | TOTAL 2018 [ha] |
|---|---|---|---|---|---|---|---|---|---|
| Urban areas | 82,448 | 0 | 0 | 0 | 0 | 0 | 0 | 0 | 0 |
| Industrial areas and infrastructure | 0 | 15,756 | 0 | 0 | 0 | 0 | 0 | 0 | 0 |
| Non-irrigated arable land | 7142 | 4134 | 245,121 | 2954 | 2361 | 1253 | 0 | 0 | 25,608 |
| Irrigated arable land | 299 | 334 | 1952 | 21,994 | 392 | 0 | 0 | 0 | 383 |
| Permanent crops | 0 | 69 | 3 | 0 | 423,500 | 3 | 0 | 0 | 0 |
| Grassland and pastures | 1309 | 3291 | 1586 | 495 | 3543 | 115,155 | 0 | 0 | 13,350 |
| Forests | 1261 | 822 | 214 | 47 | 73 | 3 | 378,104 | 0 | 598 |
| Other land cover | 0 | 0 | 0 | 0 | 0 | 0 | 0 | 6895 | 0 |
| **TOTAL 2030 [ha]** | 0 | 0 | 0 | 0 | 0 | 0 | 0 | 0 | **0** |
| Net gain/loss [ha] | **92,459** | **24,406** | **248,876** | **25,490** | **429,869** | **116,414** | **378,104** | **6895** | **39,939** |
| Net gain/loss [%] | +10,011 | +8650 | −39,697 | +136 | +6294 | −22,315 | −3018 | 0 | +39,939 |
| Simple annual rate [%] | +12.1422 | +54.8997 | −13.7563 | +0.5364 | +1.4859 | −16.0853 | −0.7918 | 0 | / |

Even in this case, the scenario is developed within a trend-based context, so that urban sprawl is still an important driver of LUC, with an increase in the urban and industrial fabric, respectively, of $+12.1\%$ and $+54.9\%$ by the end of the simulation, mostly replacing agricultural land, i.e., non-irrigated arable land ($-13.8\%$ by the end of the simulation).

The positive past trend of permanent crops is in this case reduced by the supposed spread of energy crops ($+1.5\%$ by the end of the simulation). Similarly, compared to the other two scenarios, no important change would affect irrigated arable land and forests would maintain the same trend with a little decline ($-0.8\%$ by the end of the simulation).

Ultimately, even in this case, all the pressure of human activities would lead to a reduction in grasslands ($-16.1\%$ by the end of the simulation) because of a relocation of

agriculture, commonly less competitive than secondary and tertiary sectors, but more so than grass-vegetated non-used land.

### 3.3. The Ecosystem Services Assessment Results

Figures 4 and 5 show the provision of the studied ESs in the different modeled scenarios. Both for the sediment-delivery-ratio service and for carbon storage and sequestration, the current state (land use 2018), shows higher values than the simulated scenarios by 2030. The future landscape development scenarios will probably result in a decline of environmental conditions.

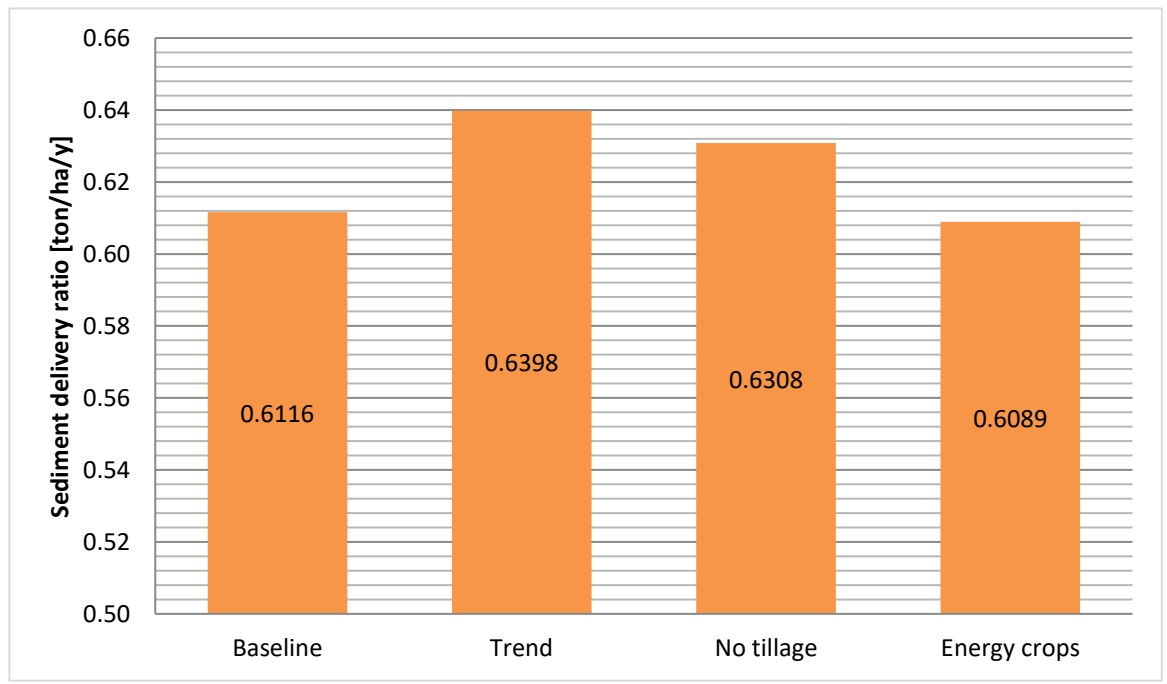

**Figure 4.** The sediment delivery ratio service assessment.

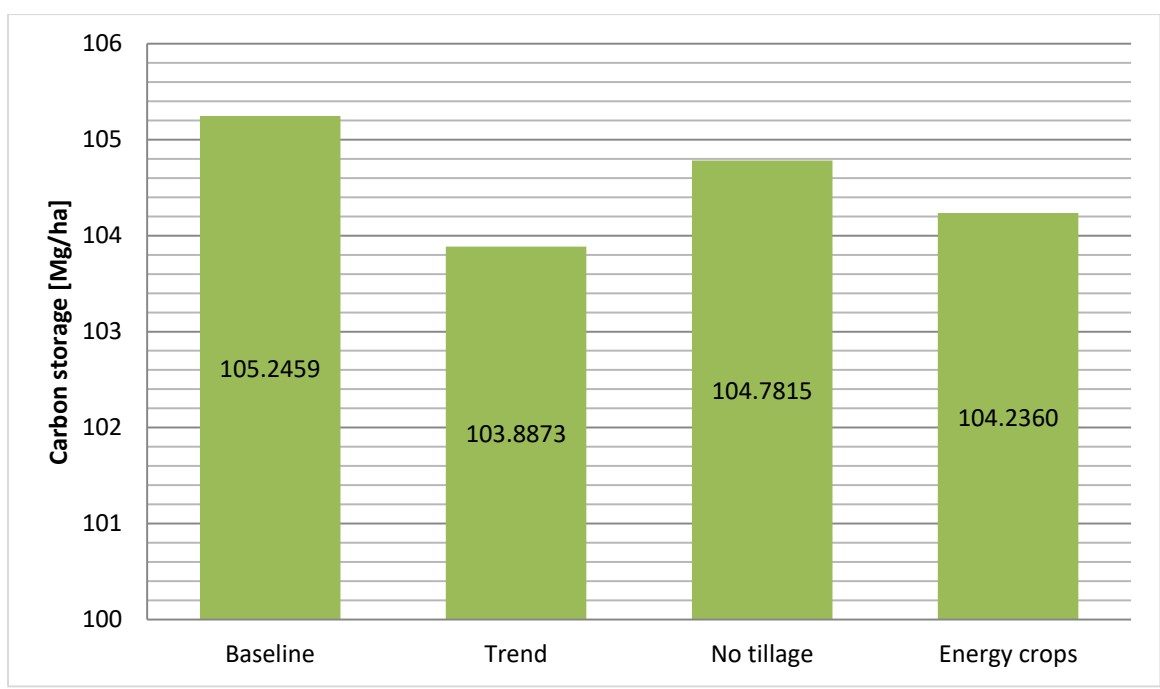

**Figure 5.** The carbon storage and sequestration service assessment.

In the trend scenario, the pressures related to the LUC (with the urban increase and the agricultural pressure on grassland) will led to a general deterioration of ecosystem value, at least as regards the considered ESs. It is the worst scenario in terms of the ES assessment. Regarding the SDR service, the increase in artificial surfaces, under the condition of impervious rainfall, will determine an increase in rainfall runoff, intensifying the erosive potential of water flow for downstream locations [76]. A large proportion of urban soil surfaces are sealed, severing the pathway between the soil surface and groundwater. The soils' contribution to flow regulation and water storage is limited, leading to increases in run-off quantity [77–79]. Similarly, agricultural land is generally more susceptible to erosion than grassland, because the canopy of cultivated plants is generally lower than that of spontaneous vegetation, so that their cover effect is limited in most cases [58]. Moreover, tillage and other agricultural practices seasonally impose a certain amount of bare soil [54]. Regarding the CSS service, artificial land cover classes are those showing the lowest carbon stocks [75,80].

In the no-tillage scenario, the results show a moderate decrease in ES values only in terms of carbon storage and sequestration service: compared to LU 2018, the scenario presents a slight reduction in CSS by 2030 (−4%), followed by the trend (−26.9%) and energy-crop scenarios (−34.5%).

In the energy-crop scenario, the sediment-delivery-ratio values are substantially comparable to those of the baseline/current state LU 2018. The ECS would be less impactful as regards soil erosion, even showing an improvement by 2030 (−3.6%), followed by no tillage (+1.2%) and trend (+17.5%).

## 4. Discussion

The first part of the present work was intended to investigate the past landscape transformations due to the effect of LUCs and to formulate hypotheses regarding future scenarios, in compliance with EU agricultural policies. Our findings highlighted urban sprawl as the main driver of past LUC. Urban sprawl mostly replaced agricultural land, especially in suburban locations, because of the expansion of existing settlements. Such a phenomenon was more intense in the valleys, where soils are normally more fertile and easily workable, and particularly in the Volturno plain, in which most of Naples's metropolitan area is located. In view of the above, to give a realistic interpretation of regional dynamics, the three scenarios proposed in this study are trend-based; therefore, each of them keeps the same general tendency in terms of urban development to the detriment of agriculture, which in turn tends to shift to locations previously covered by grassland, though with differences linked to the different hypothesized policies. As a matter of fact, while TrS tends to concentrate LUCs in hotspots, showing a more compact pattern mostly located in flatlands, in the NTS and ECS, LUCs appear more dispersed, affecting even the hilly territories of Sannio and Irpinia in the eastern part of the region, which is more suited to extensive agriculture. In particular, LUCs in the ECS are clearly shifted to the east, as a consequence of the hypothesized selection of energy crops on arable lands (Figure 3). The regional dynamics described are substantially in line with what has been happening recently in Italy, and in Europe generally [81]. In accordance with the marginal role which the agricultural sector has in the economy (and GDP) of developed countries [82,83], in the Campania region the recognition of LUCs also shows a clearly declining trend: less soil devoted to agricultural uses, but with high-income-crop intensification. A meaningful difference between the general tendency at the national and European level and Campania's regional dynamics is that, at least in the period examined, the forest surface showed a slight contraction, while on a national and European scale the expansion of forests due to the abandonment of agriculture, especially in marginal mountain areas, has become a consolidated reality in recent decades [81]. In this regard, it must be said that Campania has been the scene of several wildfires, being for years among the most affected Italian regions, together with Sicily, Sardinia, and Calabria, which have by themselves been the location for almost the totality of fire events in the country [19,84].

The results show, moreover, that both LUCs examined in the past and those hypothesized for the future are closely related to a worsening of environmental conditions. In such a context, it is even more essential to pursue the goals of sustainable development through active policies, such as those investigated by the present study, aimed at compensating for, or at least mitigating, the negative environmental impacts of LUC.

In terms of the possible environmental impact related to LUCs, the ecosystem-services approach was used to compare scenarios. The present paper focused on two specific ecosystem services that are closely connected with agro-forestry landscape management: erosion-risk mitigation and carbon storage and sequestration. The most evident concern emerging from this study is that no case simulations show an improvement of the analyzed ESs by 2030: the TrS is always the worst in terms of ES value. Regarding the other scenarios, the no-tillage scenario showed better performance in relation to carbon storage than SDR, while the energy-crop scenario led to the opposite result. The NTS scenario supposes an increase in arable lands (which is quite unlikely in view of the current trend) even on untilled land, which is mostly grassland. Conservative tillage would determine a general increase in the carbon stocks of all arable land, and this would evidently offset the loss of grassland, which is normally a higher carbon stock than any agricultural land. Regarding the energy-crop scenario, the increase in land devoted to these crops (mostly at the detriment of arable land, then of grassland) would have a positive impact on soil erosion, due to the very high canopy of energy crops, whose reference species in this case was *A. donax*. Moreover, energy crops generally require very low soil tillage (substantially comparable to no-tillage); thus, their C factor is much lower than any other agricultural land and just slightly higher than that of grassland (Table 4, Section 2.4.3). As regards carbon storage, although energy crops were considered a significant carbon stock, the overall content of organic matter in the ECS scenario is lower than that in the no-tillage one. Although this study focuses on only two ESs, it is reasonable to think that the scenarios, based on trend analysis, would result in a worsening of many other environmental indicators, given the nature of LUCs related more to the forms of use and cultivation practices than to changes among distinctly different classes. The increase in agricultural area to the detriment of natural grass-vegetated land would generally ensure worsening environmental outcomes. The land-management factor is strategic in landscape planning and management. In the evaluation of ESs, or generally of environmental indicators, the analysis of agricultural land-management practices can lead to assessments of value different from those obtained from land-use analysis only. In other words, the consideration of land management can refine the results of ES mapping so much that it can reveal situations even in contrast with expectations based on a linear logic. The methodology presented here has to be implemented in this sense; the model for evaluating carbon sequestration is based on a pre-determined amount of carbon stock associated with different land-cover types, as suggested in the "tier 1" methodology proposed by the 2006 IPCC Guidelines for National Greenhouse Gas Inventories. Changes in carbon sequestration were detected only in the case of LUC. Indeed, yearly carbon stocks are variable, even in the absence of LUC, depending on often difficult-to-consider factors, such as climate conditions and the age of species contributing to $CO_2$ fixation, and depending on site-specific conditions. Better results can be attained by making a further subdivision of land-use classes in accordance with the main vegetation species, elevation, plants' age, and forest-management factors. Another limitation of the methodology presented pertains to the SDR model, which does not consider the P-factor within the soil-loss equation [54]. For example, terracing by dry stone walls is a landscape trademark of some areas of the region, such as the Amalfi Coast and the islands, and it is an ancient technique which plays a deep role in reducing soil loss. A finer tuning of such models is hard to obtain at the regional scale, mostly due to a lack of data; however, even this is beyond the scope of the present paper.

Specific investigations, at a greater level of detail, will have to be carried out in the next steps of this work to support decision makers and the sharing and transferring of information to stakeholders. Specifically, the next steps of this work will be:

- The implementation and the validation of the proposed LUC scenario-building process and outcomes, by means of uncertainty and sensitivity analyses;
- ES-analysis implementation and integration with other ES types, closely related to biodiversity (habitat refugium and biodiversity; habitat risk; pollination; etc.) and landscape analysis (landscape metrics);
- The implementation of knowledge regarding tillage practices, integrating it with other factors, such as the inputs of agricultural production, rates of biomass removal from fields, and forest management.

The landscape-planning-process results are very complex, as one must consider different priorities, different social-economic and environmental components, different stakeholders and people involved, and different scales of interventions and of related effects, which sometimes go far beyond the boundaries of the case study. For example, climate-change mitigation is a very touchy topic from a global point of view, but it is often less appealing to local communities, which are rather more sensitive to aspects whose effect is more evident and immediately measurable [35]. Therefore, in decision making, it becomes strategic not only to respond to the requests of local communities, like in the case study related to erosion mitigation, but to consider global perspectives as well, in order to implement an effective plan of regional development. The present work, starting from past LUC analysis, mainly relates to urban spawl and the decrease in agriculture, builds possible LUC scenarios related to EU agricultural policies, and assesses specific ecosystem services, as a support to help mitigate negative environmental impacts. The study contributes to the debate in the literature on the integration of land use/landscape and ecological planning methods, focusing on the interactions among the physical and functional components of landscape. Based on this holistic approach, the definition of the driving forces, spatial modelling, impact assessment by means of the ecosystem services, and GIS-technique support, together with the creation of increasingly articulated and shared databases, are important and strategic measures in landscape planning and management, as a support for decision makers and stakeholders.

## 5. Conclusions

The present paper is framed within the field of integrating land-use modelling and ESs in decision-making processes. In particular, this work is focused on evaluating the effect of land use and land management on soil erosion and carbon-storage capacity, in a context of three different scenarios developed under alternative policies. The proposed methodology intends to suggest an integrated approach to spatial analysis, combining land-use modelling with the analysis of ESs, to provide a wide view on the matter and to make a methodological contribution to our understanding of erosion dynamics linked to LUCs. First, land-use modelling allows us to identify the areas undergoing LUCs so their eventual hotspots can be known—for example, to understand where a stronger competition between different land uses and different stakeholder' interests might occur. Secondly, a new land-cover type was introduced as a landscape element, opening the possibility of evaluating even new crops and local economy development models. Finally, in relation to agricultural uses, ESs were mapped considering aspects of land management, rather than relying on flat values associated with each land-cover type. In such a way, it was possible to obtain greater accuracy in the analysis of regional dynamics. Our findings indicate that regional development in line with past trends will lead to further land degradation. This underlines the necessity of a rational and forward-looking regional planning process. Particularly, the study confirms how the expansion of built-up areas is the main driver of land degradation in the Campania region, whose soil consumption is one the highest among all Italian regions. Thus, it seems right to highlight the importance of reducing urban sprawl by specific building plans aimed at achieving the goal of no net land acquisition, such as urban renewal and re-naturalization of dismissed artificial surfaces. Moreover, the analysis of different scenarios indicates that the measures for improving agroecosystems studied can bring the expected benefits only on the condition that they

are not the driver of important LUCs and that the naturally grass-vegetated land, which has generally a higher ecosystem value, is respected. In other words, it is important that changes in crop patterns or agricultural practices do not cause a deep alteration of the landscape arrangement; therefore, eventual subsidies should be paid only for surfaces already allocated to agricultural use.

**Supplementary Materials:** The following supporting information can be downloaded at: https://www.mdpi.com/article/10.3390/land12101865/s1, Table S1: Land-use map classes, CLC classes correspondence and Conversion elasticity. Table S2: The conversion matrix representing the allowed changes between different land-cover types.

**Author Contributions:** The authors contributed in the following ways: (E.C.) Conceptualization, Methodology, Software, Data curation, Writing—Original draft preparation, Funding acquisition; (P.F.R.) Conceptualization, Methodology, Software, Data curation, Writing—Original draft preparation; (E.S.d.P.) Investigation; Data curation; (S.P.) Conceptualization, Methodology, Supervision, Funding acquisition, Writing—Original draft preparation. All authors have read and agreed to the published version of the manuscript.

**Funding:** This research was funded by the Regione Campania Italy (Rural Development Program for 2014–2020 of Campania Region), under the following projects: "GATES" CUP G81J21000010007 and "SEI CON il Vesuvio" CUP G61B2000090007 (assigned to E. Cervelli).

**Data Availability Statement:** Data are available upon requests.

**Conflicts of Interest:** The authors declare no conflict of interest. The funders had no role in the design of the study; in the collection, analyses, or interpretation of data; in the writing of the manuscript, or in the decision to publish the results.

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
