# Peer review of "Land Use Change Scenario Building Combining Agricultural Development Policies, Landscape-Planning Approaches, and Ecosystem Service Assessment: A Case Study from the Campania Region (Italy)"

_land, doi:10.3390/land12101865_

Round 1

Reviewer 1 Report

Comments for the Manuscript “Land use change scenarios building combining agricultural development policies, landscape planning approaches and ecosystem services assessment. The Campania Region (Italy) case-study” (Manuscript Number: land-2532992) 

General comments:

Human activities are the primary cause of gradually changes in terrestrial landscapes. These extensive landscape changes can, in turn, lead to alterations in ecosystem services, ultimately impacting human habitation and livelihoods. This manuscript (land-2532992), based on LUC and incorporating ecosystem services assessment, attempts to establish an intuitive, simple and effective research framework to inform sustainable development and landscape planning. The background and significance of the study is detailed description in the manuscript, and the methodological steps of the study are very detailed and the overall content of the submitted manuscript is good, but there are still some suggestions for authors to consider.

I suggest a minor revise.

Specific comments

1.      It is recommended that authors refine the title, if it could be replaced with “ : A case study in Campania Region (Italy)” ?.

2.      Line 149 and 151, CO2, line 186 km2? Is it a writing error or a journal system submission problem? Please note this in the full text.

3.      Line 185 Fig. 1, line 213, (Figure 2), The author has used two different forms, suggesting a need for uniformity.

4.      The text above the scale in Fig. 1 is difficult to read, so please enlarge it if it is important, or suggest deleting it if it is not essential to express.

5.      It is recommended that the formulas cited in the article be numbered.

6.      It is recommended that a scale be added to Fig. 3.

7.      Figure 3. The Sediment Delivery Ratio service assessment. should be Figure 4, incorrectly numbered. The corresponding subsequent serial numbers should also be corrected.

8.      The conclusion section suggests a refinement to summarize the research and results for the study. As well as the results in the abstract, it is suggested that some data results can be put to support it.

Comments for the Manuscript “Land use change scenarios building combining agricultural development policies, landscape planning approaches and ecosystem services assessment.

The Campania Region (Italy) case-study” (Manuscript Number: land-2532992)

General comments:

Human activities are the primary cause of gradually changes in terrestrial landscapes. These extensive landscape changes can, in turn, lead to alterations in ecosystem services, ultimately impacting human habitation and livelihoods. This manuscript (land-2532992), based on LUC and incorporating ecosystem services assessment, attempts to establish an intuitive, simple and effective research framework to inform sustainable development and landscape planning. The background and significance of the study is detailed description in the manuscript, and the methodological steps of the study are very detailed and the overall content of the submitted manuscript is good, but there are still some suggestions for authors to consider.

I suggest a minor revise.

Specific comments

1.      It is recommended that authors refine the title, if it could be replaced with “ : A case study in Campania Region (Italy)” ?.

2.      Line 149 and 151, CO2, line 186 km2? Is it a writing error or a journal system submission problem? Please note this in the full text.

3.      Line 185 Fig. 1, line 213, (Figure 2), The author has used two different forms, suggesting a need for uniformity.

4.      The text above the scale in Fig. 1 is difficult to read, so please enlarge it if it is important, or suggest deleting it if it is not essential to express.

5.      It is recommended that the formulas cited in the article be numbered.

6.      It is recommended that a scale be added to Fig. 3.

7.      Figure 3. The Sediment Delivery Ratio service assessment. should be Figure 4, incorrectly numbered. The corresponding subsequent serial numbers should also be corrected.

8.      The conclusion section suggests a refinement to summarize the research and results for the study. As well as the results in the abstract, it is suggested that some data results can be put to support it.

Author Response

Comments for the Manuscript “Land use change scenarios building combining agricultural development policies, landscape planning approaches and ecosystem services assessment.The Campania Region (Italy) case-study” (Manuscript Number: land-2532992)

Reviewer #1

>> We want to thank the referee for his/her help in the general improvement of the paper and for important issues arisen to be addressed in the near future. In red we have highlighted our answers to the reviewer

General comments:

Human activities are the primary cause of gradually changes in terrestrial landscapes. These extensive landscape changes can, in turn, lead to alterations in ecosystem services, ultimately impacting human habitation and livelihoods. This manuscript (land-2532992), based on LUC and incorporating ecosystem services assessment, attempts to establish an intuitive, simple and effective research framework to inform sustainable development and landscape planning. The background and significance of the study is detailed description in the manuscript, and the methodological steps of the study are very detailed and the overall content of the submitted manuscript is good, but there are still some suggestions for authors to consider.

I suggest a minor revise.

Specific comments

  1. It is recommended that authors refine the title, if it could be replaced with “ : A case study in Campania Region (Italy)” ?.

>>We thank the Reviewer for this comment. We have replaced the sentence:

Land use change scenarios building combining agricultural development policies, landscape planning approaches and ecosystem services assessment. A case study in Campania Region (Italy)

  1. Line 149 and 151, CO2, line 186 km2? Is it a writing error or a journal system submission problem? Please note this in the full text.

>>We thank the Reviewer for arising these typos. We corrected the reported writing errors and we checked the full text.

  1. Line 185 Fig. 1, line 213, (Figure 2), The author has used two different forms, suggesting a need for uniformity.

>>We thank the Reviewer for arising the error. We uniformed the forms and we checked the full text.

  1. The text above the scale in Fig. 1 is difficult to read, so please enlarge it if it is important, or suggest deleting it if it is not essential to express.

>>We thank the Referee, we have enlarged the text in Figure 1.

  1. It is recommended that the formulas cited in the article be numbered.

>>We thank the Referee, we numbered the formulas.

  1. It is recommended that a scale be added to Fig. 3.

>>We thank the Reviewer for arising the lack. We added it.

  1. Figure 3. The Sediment Delivery Ratio service assessment. should be Figure 4, incorrectly numbered. The corresponding subsequent serial numbers should also be corrected.

>>We thank the Reviewer for arising the error. We re-numbered the figures and we checked the text.

  1. The conclusion section suggests a refinement to summarize the research and results for the study. As well as the results in the abstract, it is suggested that some data results can be put to support it.

>>We thank the Reviewer for this observation. We have reviewed the Conclusions section:

Lines 770-779: “The present paper is framed in the field of integrating land-use modelling and ESs within decision-making processes. In particular, the work is focused on evaluating the effect of land use and land management on soil erosion and carbon storage capacity, in a context of three different scenarios developed under alternative policies. Considering the complexity of studied ESs trade-offs, the present study is not aimed at the identification of a precise policy line, but rather to give a methodological contribution to the topic, although coming to important considerations about Campania’s regional dynamics. The proposed methodology wants to suggest an integrated approach to spatial analysis, combining land-use modelling with the analysis of ESs, to provide a wide view on the matter and to give a methodological contribution to the erosion dynamics linked to the LUCs ones. First, land-use modelling let to identify the undergoing LUCs areas so their eventual hot-spots: for example, to understand where a stronger competition between different land uses and different stakeholder' interests might occur. Secondly, a new land cover type was introduced as a landscape element, opening the possibility of evaluating even new crops and local economy development models. Finally, in relation to agricultural uses, ESs were mapped considering aspects of land management, rather than relying on flat values associated with each land-cover type. In such a way it was possible to obtain a greater accuracy in the analysis of regional dynamics since results revealed aspects partially mismatching expectations. Findings point out that a regional development in line with past trend will lead to further land degradation, aggravating an already troubling environmental and social situation. This underlines the necessity of a rational and forward-looking regional planning process. Particularly, the study confirms how expansion of built-up areas is the main driver of land degradation in the Campania region, whose soil consumption is one the highest among all Italian regions (Munafò, 2019). Thus, it seems right to highlight the importance of reducing urban sprawl by specific building plans aimed at achieving the goal of no net land take, such as urban renewal and re-naturalization of dismissed artificial surfaces. Moreover, the analysis of different scenarios points out that the studied measures for improving agroecosystems can bring the expected benefits only on the condition that they are not the driver of important LUCs and respecting the naturally grass-vegetated land, which has generally a higher ecosystem value. In other words, it is important that changes within crop pattern or agricultural practices don’t cause a deep alteration of landscape arrangement, therefore eventual subsidies should be paid only for surfaces already allocated to agricultural use. From the local to global scale, any spatial policy should be based on a deep knowledge of lands and a complete comprehension of social, economic, and environmental issues.

Reviewer 2 Report

General comments

The paper proposes a framework of action including different landscape planning and ecological approaches. The paper uses several metrics, ranging from those of land use and ecosystem services. Considering the relevance of this topic, this is an excellent research work. However, in many places, it is very loosely written which needs some modifications. The chosen methodology perfectly serves the analytical purposes and could be used in similar research in other countries as well. Results are good and clearly explained, and give some new contributions to the current body of knowledge in the examined area. The organization of the manuscript and presentation of the data and results need some improvement. Below I listed major and minor comments for the authors’ consideration in the hopes that it will help the authors improve their manuscript.

Specific comments:

-        Re-arrange the abstract. Line 25: Recommendation or implications should be at the end.

-        Difficult to catch up with the introduction section. It must be tightened up to give a clear message. Make it succinct and precise. Please, summarize it. In addition, harmonize the abbreviations in the manuscript or add a section of abbreviations and acronyms.

As the EU has many policies including Common Agricultural Policies, LULUCF regulations, etc. which one did you consider for this study? I suggest you follow this approach while writing your introduction: 1. Start broadly and then narrow down 2. State clearly the aim and importance 3. Cite thoroughly but not excessively, 4. Clearly state either your hypothesis or research question, and 5. Consider giving an overview of the paper, 6. Short policy implications

Line 26-27: To what extent?

Line 48: It is “Common” instead of “Community”

-        Fig 2. Increase the resolution. It is not clear for the boxes.

-        Table 2: * and – explain it in the caption. Why? Some tables need to be in the supplementary materials.

-        The discussion section must be concise and scientifically based. i. Observations from this study, ii. Comparative analysis, iii. Findings: Implications and explanations, iv. Strengths and limitations, v. Summary, recommendations, and future research opportunities.

-        The conclusion should be one paragraph. Please, modify it and avoid references in this section.

This manuscript needs moderate editing in some sections. 

Author Response

Comments for the Manuscript “Land use change scenarios building combining agricultural development policies, landscape planning approaches and ecosystem services assessment. The Campania Region (Italy) case-study” (Manuscript Number: land-2532992)

Reviewer #2

>>We want to thank the referee for his/her help in the general improvement of the paper. In red we have highlighted our answers to the reviewer

General comments:

The paper proposes a framework of action including different landscape planning and ecological approaches. The paper uses several metrics, ranging from those of land use and ecosystem services. Considering the relevance of this topic, this is an excellent research work. However, in many places, it is very loosely written which needs some modifications. The chosen methodology perfectly serves the analytical purposes and could be used in similar research in other countries as well. Results are good and clearly explained, and give some new contributions to the current body of knowledge in the examined area. 

>>We thank the Reviewer very much.

The organization of the manuscript and presentation of the data and results need some improvement. Below I listed major and minor comments for the authors’ consideration in the hopes that it will help the authors improve their manuscript.

>>Thanks to the reviewer. According to his/her comments we have improved the manuscript.

Specific comments:

  1. Re-arrange the abstract. Line 25: Recommendation or implications should be at the end.

 >>We thank the Reviewer for this observation. We have re-placed the sentence and re-arranged the abstract:

Lines 18-31: In the last two centuries, land use change (LUC) has been the most important direct changes driver for terrestrial ecosystems. To contrast the consequent ecosystems degradation, forward-looking spatial policies and target landscape and land-use planning processes, promoting a sustainable land use change from a sustainability perspective, are needed. The present paper proposes a framework of action including different landscape planning and ecological approaches: from the spatial modelling to recognize the LUC and build different scenarios, to the ecosystem services (ESs) assessment to evaluate the possible environmental impacts. Three different scenarios were built: Trend, No-Tillage and Energy crops. The Sediment Delivery Ratio and Carbon Storage and Sequestration ESs were assessed and compared for each scenario. Results show that a regional development in line with past trend could lead to further land degradation (with ESs value losses, in a decade, greater than 5%). Instead, the two scenarios proposed in compliance with EU policies, could bring benefits only if related to moderate LUCs and respecting the naturally grass-vegetated land. The aim of the paper is to support decision-makers and local communities into the landscape planning process. From the local to global scale, a guided and shared LUC management allows implementing sustainable development, basing on a deep knowledge of physical-environmental but also social and economic issues.”

  1. Difficult to catch up with the introduction section. It must be tightened up to give a clear message. Make it succinct and precise. Please, summarize it. In addition, harmonize the abbreviations in the manuscript or add a section of abbreviations and acronyms.  I suggest you follow this approach while writing your introduction:
    1. Start broadly and then narrow down
    2. State clearly the aim and importance
    3. Cite thoroughly but not excessively,
    4. Clearly state either your hypothesis or research question, and
    5. Consider giving an overview of the paper,
    6. Short policy implications

 >>We thank the Reviewer for his/her comments. We have followed the proposed approach in the Introduction section

Lines 36-121: “The land use change (LUC) is considered one of the most important driver of environmental degradation in the last two centuries (MEA, 2005) and negative LUC effects on the environment are increasingly ascertainable: habitats endangerment, nutrient cycle alteration, hydrogeological risk, etc. (Agarwal et al., 2002; Foley et al., 2005). Guiding LUC, or at least mitigating its effects, is one of the global challenges that academia and international organizations, such as the local communities, are facing. Since the last decade, different landscape policies (at International, Community and National level) are promoting sustainable land use strategies relating to different interests: nature, agriculture, energy, infrastructures, housing, etc. At European level, the Common Agricultural Policy (CAP) (EC, 2018), the European Strategy for Biodiversity 2030 (EC, 2011), the Green Deal (EC, 2019), the Renewable Energy Directive I, II, III (EC, 2020b), etc., all recognize a fundamental role of landscape and its correct management to achieve a sustainable development, anyway, none of them indicate precisely how and where accomplish their respective goals (Hersperger et al., 2020). As a result, an excessive soil consumption is still the main concern linked to the lack of coordination among policies (Montanarella & Panagos, 2021). Policy makers should be able to find a synthesis between human needs and preservation of natural resources, considering local necessities and global issues (Wu, 2021).

A combination of different land-use and landscape planning approaches can be useful to achieve a sustainable development. The study of past LUC and its implications is key to build future scenarios, which allow to assess potential impact of anthropic activities on environment and human well-being. In this field, both the land-use modelling (Veldkamp and Lambin, 2001; Verburg et al., 2019) and the mapping of ecosystem services (ESs) (MEA, 2005; Costanza et al., 2017; Hernández‐Blanco et al., 2022) are valid tools supporting decision-makers in their choices. Despite the scientific community has so far recognized the need to integrate the ESs assessment in the landscape planning process (von Haaren et al., 2019), the application of integrated evaluations is still not widespread: there’s still a lack of knowledge needing to be filled, especially as regards standardization of methods and easiness of application in environmental impact assessments (Koschke et al., 2013).

The aim of the present work is to propose a framework of actions which, integrating these different landscape/ecological planning approaches, support decision-makers and local communities to achieve a sustainable development in the rural landscape.

The present paper, in compliance with the Drivers-Pressures-State-Impacts-Responses (DPSIR) model (Rapport & Friend, 1979; EC, 1999; von Haaren et al., 2019), builds different LUC scenarios according to alternative EU agricultural policies (CAP), in order to verify whether these policies are feasible at the local level and which implications they could have on the landscape. To this end, four different scenarios were compared in terms of expected ecosystem services. The study area is the Campania region (Southern Italy), that has witnessed significant LUCs in the last decades (Cervelli and Pindozzi, 2022). The present study focuses on rural areas simulating two different kind of LUCs associated with the new 2023-2027 Italian Common Agricultural Policies (CAP) aimed at the mitigation of soil erosion, in comparison with current trend of change and no changes option. Both scenarios are devoted to analyze changes of arable land pattern, in one case as a consequence of crop changes into energy crops (to meet Green Deal requirement), in the second as a consequence of cultivation techniques changes from conventional into no-tillage.

Since LUC is not the only factor affecting ecosystems degradation, in the present study, even specific landscape management implications were deepened: land management has a profound impact on the provision of ES overall, since it can improve systems efficiency and environmental quality (Caride et al., 2012; Koschke et al. 2013; Panagos et al., 2015). Accordingly, different kinds of land management may determine different outputs of ESs, supported by new and more articulated assessment methods integrating expert opinions and spatial modelling tools (Costanza et al., 2014; von Thenen et al., 2020; Marino et al., 2022). Based on these premises, the proposed framework of action aims to deepen whether, among the agricultural uses, different management policies and strategies could directly influence not only the human activities, but also the environmental impacts and ESs supply, avoiding or, at least, mitigating negative effects of LUCs.

Despite the study was conceived as a cross-cutting research, with interactions and dialogues of intermediate and final results in every phase of the research, the paper entails the following main steps:

  • The scenarios building: three different scenarios of land-use change in Campania region (southern Italy) were built, in addition to the current state analysis.
  • The selection of ESs, their quantification and the GIS-supported mapping.
  • The comparison of scenarios, in terms of possible LUC impacts.
  • The decision-makers and communities' support, by means of specific comments on scenarios.

Specifically, the present work starts from different scenarios building, identifying the possible hot spots of LUC with the use of Dyna-CLUE modelling. Then, the work focuses on two specific ES assessment by means of InVEST software, with a specific attention to landscape use and management. The sediment delivery ratio (SDR) and carbon storage and sequestration were the reference ecosystem services investigated in this study. Such choice was driven by the will of relating a typical regional-scale issue, such as the soil erosion, to a global one, which is CO2 emissions, as a key challenge of policy making.

The study-case provides also a methodological reference for a rational spatial planning, proposing a framework of action referred to integrated approaches of land-use and landscape planning. Understanding the dynamics of landscape transformation from local to global scale, helps limit or at least mitigate its negative effects. Agricultural areas can play a fundamental role in land use change planning and landscape management, both in terms of provisioning and regulating functions. The International, EU and National strategies which link LUC and environmental impacts, are useful guides at global level to pursue a sustainable development, but have to be improved at local level to find concrete application solutions and to mitigate and integrate the LUC impacts. Inside the rural areas, the degraded or marginal, or fringe areas can respond, better of other, to the challenges of a resilient land development.

  1. As the EU has many policies including Common Agricultural Policies, LULUCF regulations, etc. which one did you consider for this study?

>>According to the referee suggest we specified the policies:

Lines 75-82:The present study focuses on rural areas simulating two different kind of LUCs associated with the new 2023-2027 Italian Common Agricultural Policies (CAP) aimed at the mitigation of soil erosion, in comparison with current trend of change and no changes option. Both scenarios are devoted to analyze changes of arable land pattern, in one case as a consequence of crop changes into energy crops (to meet Green Deal requirement), in the second as a consequence of cultivation techniques changes from conventional into no-tillage.”

  1. Line 26-27: To what extent?

 >>Thank to the Reviewer, we have added the extent:

Line 26: “(with loss of value, in a decade, for ES greater than 5%).”

  1. Line 48: It is “Common” instead of “Community”

>>We thank the reviewer to have highlight the mistake.

  1. Fig 2. Increase the resolution. It is not clear for the boxes.

>>According to the referee suggest we increased the Figure 2 resolution.

  1. Table 2: * and – explain it in the caption. Why? Some tables need to be in the supplementary materials.

>>We thank the Reviewer for arising the lack. explained “*” in the caption (Annex 2) and we re-placed some table in the supplementary materials.

Annex 2 caption: “*: the asterisk "*" indicates that the LU class is foreseen (and has to be taken into account) only in one scenario: the Energy crops scenario (ECs)”.

  1. The discussion section must be concise and scientifically based.
    1. Observations from this study,
    2. Comparative analysis, iii.
    3. Findings: Implications and explanations,
    4. Strengths and limitations,
    5. Summary, recommendations, and future research opportunities.

>>According to the referee suggest we re-arranged the Discussions section:

Lines 654-766:The first part of the present work was intended to investigate the past landscape transformations due to the effect of the LUC and to formulate hypotheses of future scenarios, in compliance with EU agricultural policies. Findings highlighted the urban sprawl as the main driver of past LUC. The urban sprawl mostly replaced agricultural land, especially in suburban locations, because of the existing settlements expansion. Such a phenomenon was more intense in the valleys, where soils are normally more fertile and easily workable, and particularly in the Volturno plain, where it is located most of Naples metropolitan area. In view of the above, to give a realistic interpretation of regional dynamics, the three scenarios proposed in this study are trend-based, therefore each of them keeps the same general tendency in terms of urban development at the detriment of agriculture, which in turn tends to shift on locations previously covered by grassland, though with differences linked to the hypothesized different policies. As a matter of fact, while TrS tends to concentrate LUCs in hotspots showing a more compact pattern mostly located in flatlands, in the NTS and ECS LUCs appear more disperse, affecting even hilly territories of Sannio and Irpinia in the eastern part of the region, which is more suited for extensive agriculture. In particular, LUCs in the ECS are clearly shifted to the East, as a consequence of the supposed affirmation of energy crops on arable lands (Figure 3). The described regional dynamics are substantially in line with what has been lately happening in Italy, and generally in Europe (Munafò and Marinosci, 2018). In accordance with the marginal role which agricultural sector has in the economy (and GDP) of developed Countries (Kour et al., 2020; Velasco-Muñoz et al., 2021), also in Campania region the recognition of LUCs shows a trend clearly declining: less soil de-voted to the agricultural uses, but with the high-income crops intensification. A meaningful difference between the general tendency at national and European level and Campania’s regional dynamics is that, at least in the examined period, forest sur-face showed a slight contraction, while on a national and European scale the expansion of forests due to abandonment of agriculture, especially in marginal mountain areas, is a consolidated reality of last decades (Munafò and Marinosci, 2018). In this regard, it must be said that Campania has been the scene of several wildfires, being for years among the most affected Italian regions together with Sicily, Sardinia and Calabria, which hold by themselves almost the totality of fire events in the country (Silvestro et al., 2021; Cervelli et al., 2022). The results show, moreover, that both LUCs examined in the past and those hypothesized for the future are closely related to a worsening of environmental conditions. In such a context, it is even more essential to pursue goals of sustainable development through active policies such as those investigated by the present study, aimed at compensating or, at least, mitigating negative environmental impacts of LUC.

In terms of possible environmental impact related to LUCs, the ecosystem services approach was used to compare scenarios. The present paper focused on two specific ecosystem services, that are closely connected with the agro-forestry landscape management: the erosion risk mitigation and the carbon storage and sequestration. The most evident concern emerging from this study is that in no case simulations show an improvement of the analyzed ESs by 2030: the TrS is always the worst in terms of ESs value. About the other scenarios, the no-tillage one showed a better performance in relation to carbon storage rather than to SDR, while the energy crops scenario led to opposite results. The NTS scenario supposes an increase of arable lands (which is quite unlikely in compliance with actual trend) even on untilled land, mostly grassland. The conservative tillage would determine a general increase of carbon stocks of all arable land, and this would evidently offset the loss of grassland, which is normally a higher carbon stock than any agricultural land. About the energy crops scenario, the increase of devoted crops (mostly at the detriment of arable land, then of grassland) would have a positive impact on soil erosion, due to the very high canopy of energy crops, whose reference species was in this case A. donax. Moreover, the energy crops generally require a very low soil tillage (substantially comparable to no-tillage), thus their C factor is much lower than any other agricultural land and just slightly higher than that of grassland (Table 4, section 2.3.1.1). As regards carbon storage, despite energy crops were considered as a significant carbon stock, the overall content of organic matter in the ECS scenario is lower than the no-tillage one. Although the study focuses on only two ESs, it is reasonable to think that the scenarios, based on trend analysis, would result in a worsening of even many other environmental indicators, given the nature of LUCs related more on the forms of use and cultivation practices that on the changes among distinctly different classes. The increase of agricultural areas at the detriment of natural grass-vegetated lands would generally determine environmental worsen. The land management factor is strategic in the landscape planning and management. In the evaluation of ESs, or generally of environmental indicators, the agricultural land management practices analysis can lead to values assessments different from those obtained from only land uses analysis. In other words, the consideration of land management can refine results of ESs mapping, so much so that it can reveal situations even in contrast with expectations based on a linear logic. The presented methodology has to be implemented in this sense: the model for evaluating carbon sequestration is basically based on a pre-determined amount of carbon stock associated with different land-cover types, as suggested by the “tier 1” methodology proposed by 2006 IPCC Guidelines for National Greenhouse Gas Inventory. Changes in carbon sequestration are detected only in case of LUC. Actually, Carbon stocks are yearly variable, even in absence of LUC, depending on factors often difficult to consider, such as climate conditions and age of species contributing to CO2 fixation, and depending on site-specific conditions. Better results can be performed by making a further subdivision of land-use classes in compliance with main vegetation species, elevation, plants’ age, and forest management factors. Another limitation of the presented methodology regards SDR model, which doesn’t consider the P-factor within the soil loss equation (Wishmeier and Smith, 1978). For example, terracing by dry stone walls is a landscape trademark of some areas of the region, such as the Amalfi Coast and the islands, and it is an ancient technique which plays a deep role in reducing soil loss. A finer tuning of such models is hard to obtain at regional scale, mostly due to lack of data, however, even this is beyond the scope of the present paper.

Specific investigations, at a greater level of detail, will have to be carried out in the next work steps to support decision-makers and sharing and transferring information with the stakeholders. Specifically, next steps of the work will be:

  • The implementation and the validation of proposed LUC scenarios building process and outcomes, by means of uncertainty and sensitivity analyses.
  • The ES analysis implementation and integration with other ESs types, closely related to the biodiversity (habitat refugium and biodiversity; habitat risk; pollination; etc.) and the landscape analysis (landscape metrics).
  • The implementation of the knowledge on tillage practices, integrating them with other factors, such as inputs of agricultural production, rates of biomass removal from fields and forest management.

The landscape planning process results very complex, having to consider different priorities, different social-economic and environmental components, different stakeholders, and people involved, different scale of interventions and of related effects, which sometimes far beyond the boundaries of the case study. For example, climate change mitigation is a very touchy topic from a global point of view, but it’s often less appealing to local communities, which are rather more sensitive to aspects whose effect is more evident and immediately measurable (Shoyama and Yamagata, 2014). Therefore, into decision making, it becomes strategic not only to respond to the requests of local communities, like in the case study related to erosion mitigation, but considers even global perspectives, in order to implement an effective plan of regional development. The present work, starting from the past LUC analysis, mainly relate to the urban spawl and agriculture decrease, builds possible LUC scenarios related to EU agricultural policies and assesses specific ecosystem services, as support to mitigate negative environmental impacts. The study contributes to the literature debate on land-use / landscape and ecological planning methods integration, focusing on the interactions among the physical and functional components of landscape. Basing on the holistic approach, the definition of the driving forces, the spatial modelling, the impact assessment by means of the ecosystems services, the GIS techniques support, together with the creation of increasingly articulated and shared databases are important and strategic measures in landscape planning and management, as support for decision-makers and stakeholders.”

  1. The conclusion should be one paragraph. Please, modify it and avoid references in this section.

>>According to the referee suggest we modified the section.

Lines 770-779:The present paper is framed in the field of integrating land-use modelling and ESs within decision-making processes. In particular, the work is focused on evaluating the effect of land use and land management on soil erosion and carbon storage capacity, in a context of three different scenarios developed under alternative policies. Considering the complexity of studied ESs trade-offs, the present study is not aimed at the identification of a precise policy line, but rather to give a methodological contribution to the topic, although coming to important considerations about Campania’s regional dynamics. The proposed methodology wants to suggest an integrated approach to spatial analysis, combining land-use modelling with the analysis of ESs, to provide a wide view on the matter and to give a methodological contribution to the erosion dynamics linked to the LUCs ones. First, land-use modelling let to identify the undergoing LUCs areas so their eventual hot-spots: for example, to understand where a stronger competition between different land uses and different stakeholder' interests might occur. Secondly, a new land cover type was introduced as a landscape element, opening the possibility of evaluating even new crops and local economy development models. Finally, in relation to agricultural uses, ESs were mapped considering aspects of land management, rather than relying on flat values associated with each land-cover type. In such a way it was possible to obtain a greater accuracy in the analysis of regional dynamics since results revealed aspects partially mismatching expectations. Findings point out that a regional development in line with past trend will lead to further land degradation, aggravating an already troubling environmental and social situation. This underlines the necessity of a rational and forward-looking regional planning process. Particularly, the study confirms how expansion of built-up areas is the main driver of land degradation in the Campania region, whose soil consumption is one the highest among all Italian regions (Munafò, 2019). Thus, it seems right to highlight the importance of reducing urban sprawl by specific building plans aimed at achieving the goal of no net land take, such as urban renewal and re-naturalization of dismissed artificial surfaces. Moreover, the analysis of different scenarios points out that the studied measures for improving agroecosystems can bring the expected benefits only on the condition that they are not the driver of important LUCs and respecting the naturally grass-vegetated land, which has generally a higher ecosystem value. In other words, it is important that changes within crop pattern or agricultural practices don’t cause a deep alteration of landscape arrangement, therefore eventual subsidies should be paid only for surfaces already allocated to agricultural use. From the local to global scale, any spatial policy should be based on a deep knowledge of lands and a complete comprehension of social, economic, and environmental issues.

Round 2

Reviewer 2 Report

The authors clearly addressed all the comments. This paper can be published in Land. Keep it up.